# Timing of whole genome duplication is associated with tumor-specific MHC-II depletion in serous ovarian cancer

Nikki L. Burdett [1,2,3], Madelynne O. Willis [1], Ahwan Pandey [1], Laura Twomey[1], Sara Alaei[1,4], Australian Ovarian Cancer Study Group*, David D. L. Bowtell[1,2] & Elizabeth L. Christie [1,2] ✉

Whole genome duplication is frequently observed in cancer, and its prevalence in our prior analysis of end-stage, homologous recombination deficient high grade serous ovarian cancer (almost 80% of samples) supports the notion that whole genome duplication provides a fitness advantage under the selection pressure of therapy. Here, we therefore aim to identify potential therapeutic vulnerabilities in primary high grade serous ovarian cancer with whole genome duplication by assessing differentially expressed genes and pathways in 79 samples. We observe that MHC-II expression is lowest in tumors which have acquired whole genome duplication early in tumor evolution, and further demonstrate that reduced MHC-II expression occurs in subsets of tumor cells rather than in canonical antigen-presenting cells. Early whole genome duplication is also associated with worse patient survival outcomes. Our results suggest an association between the timing of whole genome duplication, MHC-II expression and clinical outcome in high grade serous ovarian cancer that warrants further investigation for therapeutic targeting.

Whole genome duplication (WGD) is frequently observed in many cancer types, including high grade serous ovarian cancer (HGSC)[1,2]. WGD serves as a buffer to tolerate acquisition of deleterious mutations and consequently is positively selected for under certain conditions, despite a fitness cost to cells[3]. WGD has been observed in 50–60% of primary ovarian cancers[4,5], yet we observed a higher proportion in our study of end-stage homologous recombination (HR) deficient HGSC (79.6% of tumours), suggesting that WGD continues to be acquired after diagnosis[6]. There are multiple genomic events that occur in HGSC which can predispose toward polyploidy and WGD such as *TP53* mutations, and *PIK3CA* and *CCNE1* alterations[1,7–9]. Both its frequency and continued acquisition under the selection pressure of therapy indicate that WGD is advantageous in HGSC[1,4,10].

HGSC confers a poor prognosis, with a 5-year survival approaching 50%[11–13], and is rarely cured once diagnosed, even in those with a good initial response to treatment. Our previous work starkly illustrated that the mechanisms of resistance are diverse[6], and targeted treatments do not exist for many of these in mainstream clinical practice. New treatments have been sought for HGSC, however the response to immune checkpoint inhibitor (ICI) therapy has been poor, despite evidence that the immune milieu has an important role in ovarian cancer control and survival[14–16]. This may be due to a lack of tumour or stromal immune cell infiltration, and specific mechanisms of immune escape[17]. While immune escape related to reduced neoantigen presentation has been largely described with regard to MHC-I in HGSC and other cancer types[17–19], there is increasing suggestion that MHC-II expression also plays a prominent role[20]. Canonical antigen presenting cells, including macrophages, dendritic cells and B cells, as well as cancer cells, can express MHC-II and present neoantigens[20,21]. The presence of MHC-II on cancer cells can even predict response to

[1]Peter MacCallum Cancer Centre, Melbourne, VIC 3000, Australia. [2]Sir Peter MacCallum Department of Oncology, The University of Melbourne, Melbourne, VIC 3010, Australia. [3]Box Hill Hospital, Eastern Health, Box Hill, VIC 3128, Australia. [4]Australian Regenerative Medicine Institute, Monash University, Clayton, VIC 3168, Australia. *A list of authors and their affiliations appears at the end of the paper. ✉e-mail: liz.christie@petermac.org

ICI[20,22]. Additionally, across tumour types, aneuploidy and response to ICI have been linked[23].

We hypothesized that WGD might drive unique transcriptional processes in HGSC that may contribute to disease recurrence and treatment resistance. Therefore, if these could be targeted early in cancer evolution, then development of resistance could be abrogated.

In this work, we aim to elucidate the characteristics of primary HGSC which have undergone WGD that may represent therapeutic vulnerabilities, and demonstrate here an association between the timing of WGD and MHC-II expression in HGSC, as well as with clinical outcomes.

## Results

### MHC-II pathway expression is reduced in tumours with WGD

Whole genome sequencing (WGS) data from seventy-nine primary HGSC tumour samples in the International Cancer Genome Consortium (ICGC) study, described previously by Patch et al.[24], was analysed for whole genome duplication (WGD) status. WGD was defined as the tumour having >50% of the autosomal genome with a major copy number (MCN) $\geq 2$[1]. Fifty of these tumours had WGD and 29 did not. To identify genes where expression differed between cases with and without WGD, a differential gene expression (DGE) analysis was performed using the matched RNA sequencing data and a generalised linear model (GLM) (Fig. 1a). This included covariates of purity as well as gene specific copy number inferred from FACETS[25] for each sample, since by definition tumours with WGD are likely to have a greater number of copy number amplifications potentially affecting gene expression. The coefficient estimate generated by the GLM was used as the magnitude of difference in place of log fold change, since this model needed to account for per gene, per sample covariates, unlike standard DGE analysis tools, and log fold change is not determined by the model. The median tumour purity estimated by FACETS was 68.7% in tumours without WGD and 62% in tumours with WGD (Supplementary Table 1). Of the 16,375 genes input into the model, 82 were significantly upregulated and 593 significantly downregulated using a coefficient estimate threshold of ±1.5 and a $p$ value < 0.05. The higher number of down-regulated compared to up-regulated genes in tumours with WGD is likely due to the fact that the model accounts for copy number log ratio as a covariate, since by definition tumours with WGD will have higher total copy number per gene.

Interestingly, we noted that 13 MHC-II genes had a statistically significant coefficient estimate <−1.5 in tumours with WGD compared to those without. Despite these genes being located on chromosome 6, this did not appear to be driven by a consistent chromosomal arm loss occurring in the tumours with WGD (Supplementary Fig. 1a). Similarly, comparing the segment copy number there was no significant difference in the segment mean between MHC-II genes and other genes in the same region (6p21.32–6p25.3; $p = 0.09$; Supplementary Fig. 1b). In addition, by comparing the coefficient estimates from the model for all genes within the region on 6p where these genes are located, we observed that the low expression specifically affected the MHC-II genes rather than being common to all genes in the region ($p < 0.001$, Wilcoxon's test; Supplementary Fig. 1c). CD74, which encodes the invariant chain that binds to the MHC alpha and beta chains and is located on chromosome 5, also had significantly lower expression (coeff. est. −1.95, $p < 0.001$), supporting our observation that there is specific downregulation of MHC-II-related genes in the tumours with WGD and this is independent of chromosomal loci. Finally, we re-ran our model without copy number as a covariate, producing similar results (Supplementary Tables 2, 3). This indicates that the transcriptional differences observed are independent of gene specific copy number status.

We hypothesized that altered expression of a transcription factor or upstream regulator may explain this MHC-II depletion. Consistent with this notion, we found that the Class II Major Histocompatibility Complex Transactivator (CIITA) gene, known as the master-regulator of MHC-II activity[26], was also significantly downregulated in tumours with WGD (coeff. est. = −2.06, $p < 0.001$). These DGE results were then verified in the independent TCGA dataset, specifically in a cohort of 166 HGSC tumours for which complete clinical information, copy number, tumour purity and RNAseq data was available. The intersect of genes between the 2 datasets that were differentially expressed at any magnitude was assessed ($n = 689$ genes with copy number covariate, Supplementary Table 2; $n = 550$ without copy number covariate, Supplementary Table 3). In this way, CIITA and 8 of the 13 MHC-II genes were also confirmed to be significantly more lowly expressed in the WGD samples of the TCGA validation set, albeit by a smaller magnitude (Fig. 1b).

We sought determinants of the difference in CIITA expression between samples with or without WGD. Though MHC-II is constitutively expressed by canonical antigen-presenting cells, it can also be inducibly expressed by other cells including tumour cells[27–29]. CIITA expression is typically controlled by methylation of four context-specific promoters[21,30–32]. Using our previously generated methylation array data[6,24] it was only possible to identify 1 probe likely to correspond to a CIITA promoter region; however, no samples were hypermethylated at this probe. We also did not find evidence of CIITA hypermethylation in the TCGA data. Upstream signalling factors reported to regulate the expression of CIITA include IFNγ, JAK1, STAT1, IRF1 and FBXO11[21,33,34], however none of these were significantly differentially expressed across both the discovery and validation datasets. There was however a moderate correlation between IRF1 (a positive regulator of CIITA) and CIITA expression in both datasets (ICGC R = 0.53, $p < 0.001$; TCGA R = 0.51, $p < 0.001$, Spearman's test; Supplementary Fig. 2a, b).

### Pathway analysis reveals depletion of immune response in WGD

Pathway enrichment analysis was conducted using the most highly and lowly expressed genes from the DGE results for the ICGC discovery and TCGA validation cohorts using ActivePathways[35], which integrates results from multiple datasets, to identify significantly enriched Hallmark pathways. Seven downregulated and no upregulated pathways were identified as being associated with presence of WGD (Fig. 1c). Notably, the significantly depleted processes largely related to the immune response. Downregulated pathways were similar to those described in a pan-cancer analysis of differentially expressed pathways between cancers with and without WGD, which found Allograft rejection, Inflammatory response and Interferon gamma response to be downregulated in samples with WGD[3].

### Timing of WGD is associated with MHC-II gene expression

In pan-cancer studies, WGD has been described as occurring early in tumorigenesis or comparatively 'later', around the time of diagnosis[2]; WGD may also occur after diagnosis. We hypothesised that MHC-II expression and other biological processes highlighted by our pathway analysis might be affected by the timing of WGD acquisition, and therefore used the method described by Dewhurst et al. to categorise WGD as 'early' or 'late'[36]. Their method uses genomic regions with a total copy number of 2 to categorise tumours with more heterozygous regions as having undergone genome duplication before the majority of losses (early), and tumours with more homozygous regions classified as having undergone WGD after the majority of losses (late). While this is acknowledged to be heuristic, our findings were concordant with the previous analysis by Gerstung et al.[2] for the 36 ICGC samples with WGD analysed by both methods (Supplementary Fig. 2c).

Using the ICGC discovery dataset, we observed that Hallmark pathway enrichment scores clustered by timing of WGD (Fig. 2a). Clade 1, with downregulation of pathways related to immune response identified in the analysis by ActivePathways (Interferon Gamma

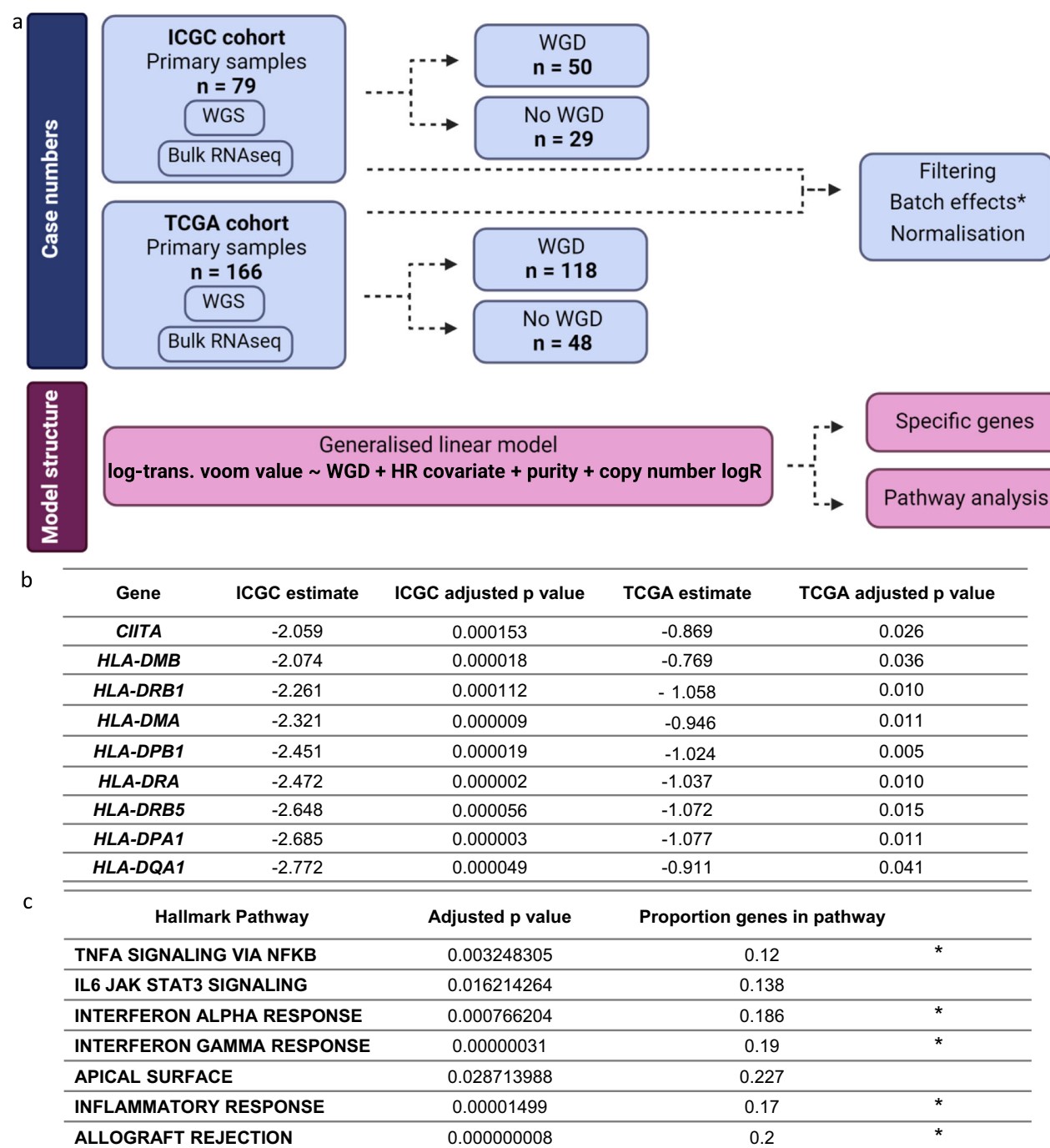

**Fig. 1 | Differential gene expression analysis workflow and key results.**
**a** Schematic demonstrating (top) case numbers for the International Cancer Genome Consortium (ICGC) discovery and The Cancer Genome Atlas (TCGA) validation cohort and workflow, and (bottom) the differential gene expression (DGE) generalised linear model structure. *The TCGA validation cohort was processed in the same manner, however batch correction was not possible since batch information is not given. Whole genome duplication (WGD), homologous recombination deficiency (HRD). Created with BioRender.com. **b** Key DGE results derived from generalised linear model for *CIITA* and MHC-II related genes with significantly lower expression in both the discovery and validation cohorts in WGD samples.
**c** Overrepresented pathways derived by integrating discovery and validation DGE results through ActivePathways[35] (ranked hypergeometric test) using the differentially expressed geness with significantly lower expression in WGD samples. Asterisks denote pathways involved in immune response. Source data are provided as a Source Data file and Supplementary Table 2. *P* values are adjusted for multiple comparisons.

response, Interferon Alpha response, Inflammatory response, Allograft rejection, and TNF signalling via NF-КB), was enriched for tumours with early WGD ($p = 0.04$, Chi squared test). This pattern of enrichment was similar in the TCGA dataset but was not statistically significant ($p = 0.14$, Chi squared test; Supplementary Table 4). As HGSC molecular subtypes are associated with the extent of epithelial and stromal immune cell infiltration and patient response to treatment[37], we also annotated each sample for these. Notably, C5 (proliferative) samples, which are immune depleted[38], exclusively fell into this clade. In keeping with this we observed that *LIN28B*, which is highly expressed in C5 tumours[39], was one of most significantly upregulated genes in WGD samples (Supplementary Tables 2, 3). In contrast, the second and third clades, enriched for tumours with late WGD and tumours without WGD respectively, generally displayed enrichment of immune related

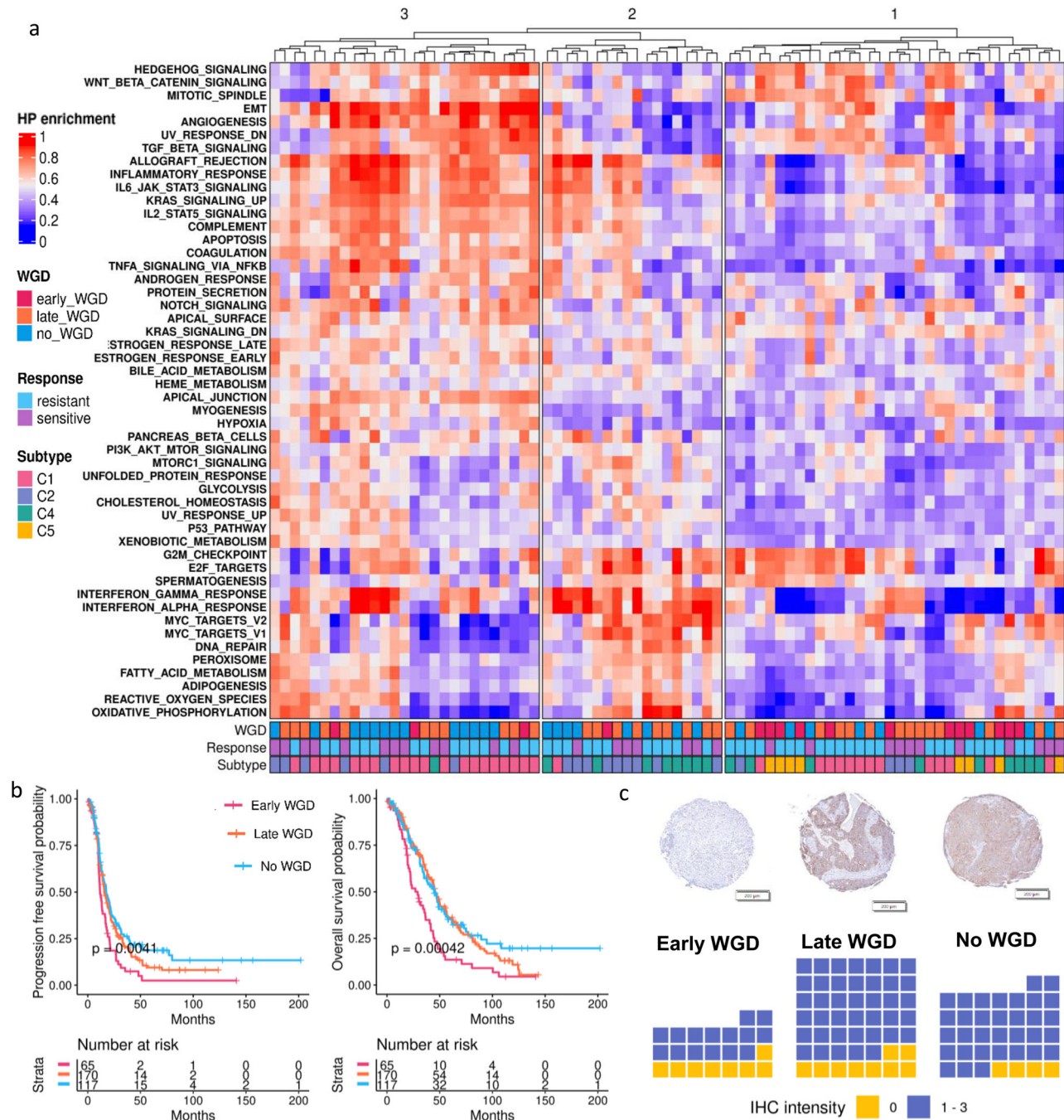

**Fig. 2 | Differences in gene expression and clinical outcomes by timing of WGD.** **a** Hierarchical clustering of Hallmark pathway enrichment scores in the discovery ICGC cohort. *n* = 79. Heatmap is annotated by timing of whole genome duplication (WGD), clinical response to first line therapy and molecular subtype (C1,C2,C4,C5). **b** Two-sided Kaplan–Meier analysis of overall survival (top) and progression free survival (bottom) by presence and timing of WGD in 352 patients (ICGC and TCGA cohorts combined). **c** Top: Representative image of core stained with anti-HLA DR + DP + DQ for each WGD category; scale bar is 200 μm. Bottom: Waffle plots depicting immunohistochemistry (IHC) results, coloured by zero (yellow) versus any (lilac) staining with anti-HLA DR + DP + DQ, separated by WGD status. Early WGD *n* = 23 cores; Late WGD *n* = 49 cores; No WGD *n* = 37 cores. Source data are provided as a Source Data file.

pathways. Clade 3 was notably enriched for the C1 (mesenchymal) subtype.

Tumours with early WGD had significantly lower expression of *CIITA* compared to those with late or no WGD (*p* = 0.04, Kruskal-Wallis test, ICGC dataset, Supplementary Fig. 2d). The same pattern was observed for the 8 MHC-II genes that had been identified as significantly differentially expressed across both the ICGC and TCGA cohorts. Expression of these genes was lowest in the tumours with early WGD, though only 4 of 8 genes were statistically significant after

adjustment for multiple testing (Supplementary Fig. 2d). Taken together, reduced *CIITA* and MHC-II gene expression was more strongly associated with early WGD, rather than just WGD generally. We hypothesized then that reduced MHC-II expression might contribute to immune evasion in patients with tumours which have undergone early WGD, and that therefore patients with early WGD might have poorer survival outcomes. In order to have sufficient power to detect survival differences by WGD timing we examined patient survival in a combined cohort (79 ICGC cases and 166 TCGA cases, plus 107

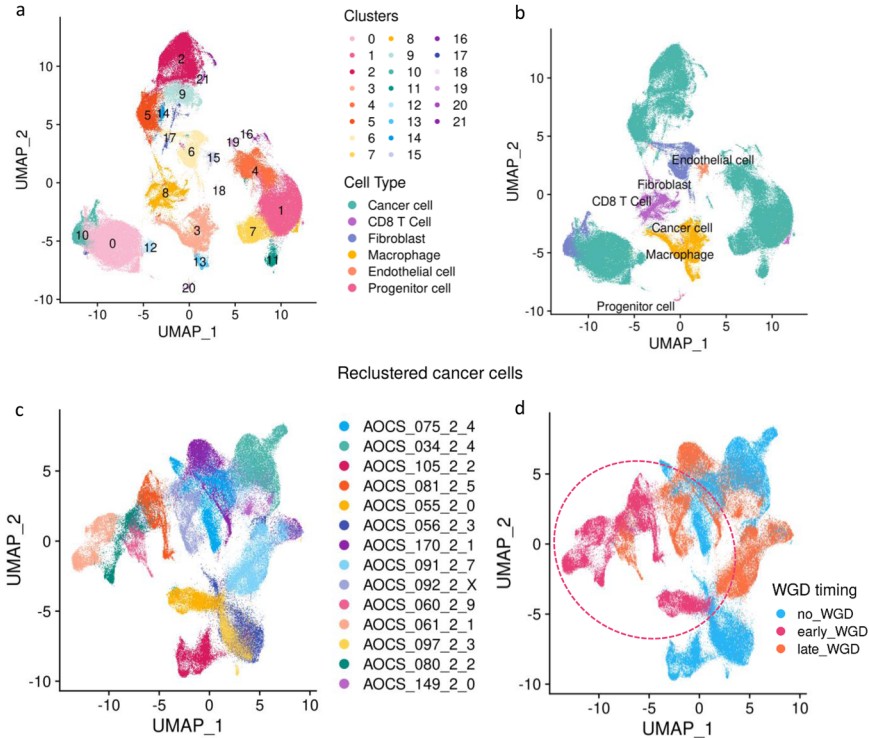

**Fig. 3 | Single nuclei RNA sequencing. a** UMAP plots of all cells coloured by Seurat clusters. **b** UMAP plots of all cells coloured by cell types. **c** UMAP of reclustered cancer cells coloured by patient. **d** UMAP of reclustered cancer cells coloured by whole genome duplication (WGD) timing (inferred per patient from bulk WGS). Ellipse highlights early WGD clustering. For **a** and **b**: Early WGD $n = 4$ patients, 33,965 nuclei; Late WGD $n = 5$ patients, 63,156 nuclei; No WGD $n = 5$ patients, 80,680 nuclei. **c**, **d** Early WGD $n = 4$ patients, 31,479 nuclei; Late WGD $n = 5$ patients, 48,685 nuclei; No WGD $n = 5$ patients, 60,602 nuclei.

additional TCGA cases for which WGS and survival information were accessible; total of 352 patients). Strikingly, both progression free survival (PFS) and overall survival (OS) were significantly worse for cases with early WGD, with little separation between those with late or no WGD (Fig. 2b). This remained significant in multivariate analysis with both age and stage at diagnosis, using Cox proportional hazards for both OS ($p = 0.002$ late WGD, $p = 0.03$ no WGD) and PFS ($p = 0.01$ late WGD, $p = 0.003$ no WGD; Supplementary Fig. 2e,f).

### Tumour-specific CIITA and MHC-II gene expression

We next sought to determine whether the reduced MHC-II expression seen in bulk sequencing data from tumour samples with WGD was due to altered expression by tumour or antigen-presenting cells. While canonical antigen-presenting cells such as dendritic cells, macrophages and B cells are typically the source of MHC-II expression, tumour cell specific MHC-II expression has been shown in ovarian cancer as well as other cancer types[28,29,40], where it predicts response to immune checkpoint inhibition[22,41]. Using CIBERSORTx immune cell imputations, we did not observe a difference in proportion of B cells, dendritic cells nor macrophages by timing of WGD (Supplementary Tables 5, 6), hence we hypothesised that differences in MHC-II expression were caused by inducible expression in tumour cells, as has been suggested in other studies[29,40]. To ascertain the cell types expressing MHC-II, we evaluated protein expression of MHC-II by immunohistochemistry (IHC) using an anti-HLA DR + DP + DQ antibody. Sixty-one of the 79 ICGC tumours were tested (13 with early WGD, 27 with late WGD and 21 without WGD). Review of the IHC by a gynaecologic pathologist determined that MHC-II was expressed in the tumour cells as well as immune cells. MHC-II IHC staining intensity was reduced in tumour cells from cases with early WGD compared to either late or no WGD ($p = 0.049$, glmm, Supplementary Fig. 3). In contrast, neither tumour-infiltrating lymphocytes nor stromal lymphocytes displayed a significant difference in MHC-II staining intensity by timing

of WGD. Overall, there were proportionally more tumours with a complete absence of MHC-II expression in the group with early WGD, yet this did not reach statistical significance ($p = 0.06$, Chi squared test, Fig. 2c).

To address which cells are expressing *CIITA* and to examine the intercellular MHC-II signalling associated with this, we performed single nuclei RNA sequencing (snRNAseq) of 5 patient samples each with either early, or late or no WGD ($n = 15$, Supplementary Table 7). For one sample, sufficient nuclei could not be extracted at the lysis step and was not processed further, hence a total of 195,712 single nuclei from tumour samples from 14 patients were sequenced and passed quality control. Following filtering and transformation, principal component analysis and UMAP analysis generated 22 clusters from 177,801 nuclei. Cells were annotated using a consensus method of cancer cell markers and marker-based annotation with ScType[42], and assessed for aneuploidy prediction using CopyKAT (Fig. 3a, b, Supplementary Table 8).

Fifteen cancer cell clusters were identified through cell type annotation. These were then re-clustered separately to assess cancer cell-specific differences (Fig. 3c, d). Annotated by WGD status, this demonstrated a macro-cluster of cancer cells from tumours with early WGD, separate from cancer cells with late or no WGD (Fig. 3d).

We confirmed that *CIITA* was expressed in proportionally fewer cancer cells within each patient sample in tumours with early WGD, compared to those with late or no WGD ($p = 0.04$, Fig. 4a). Comparison of the level of *CIITA* expression across all cancer cell clusters confirmed that *CIITA* was also significantly more lowly expressed in early WGD tumours (coeff. est. = 4.91 late WGD, $p$ value $< 0.001$, coeff. est. = 4.18 no WGD, $p$ value $< 0.001$, glmm; Fig. 4b). This was also reflected in the proportions of cancer cells expressing MHC-II genes across and within cases (Fig. 4c, d, Supplementary Fig. 4). Further analysis revealed that *CIITA* expression levels in cancer cells

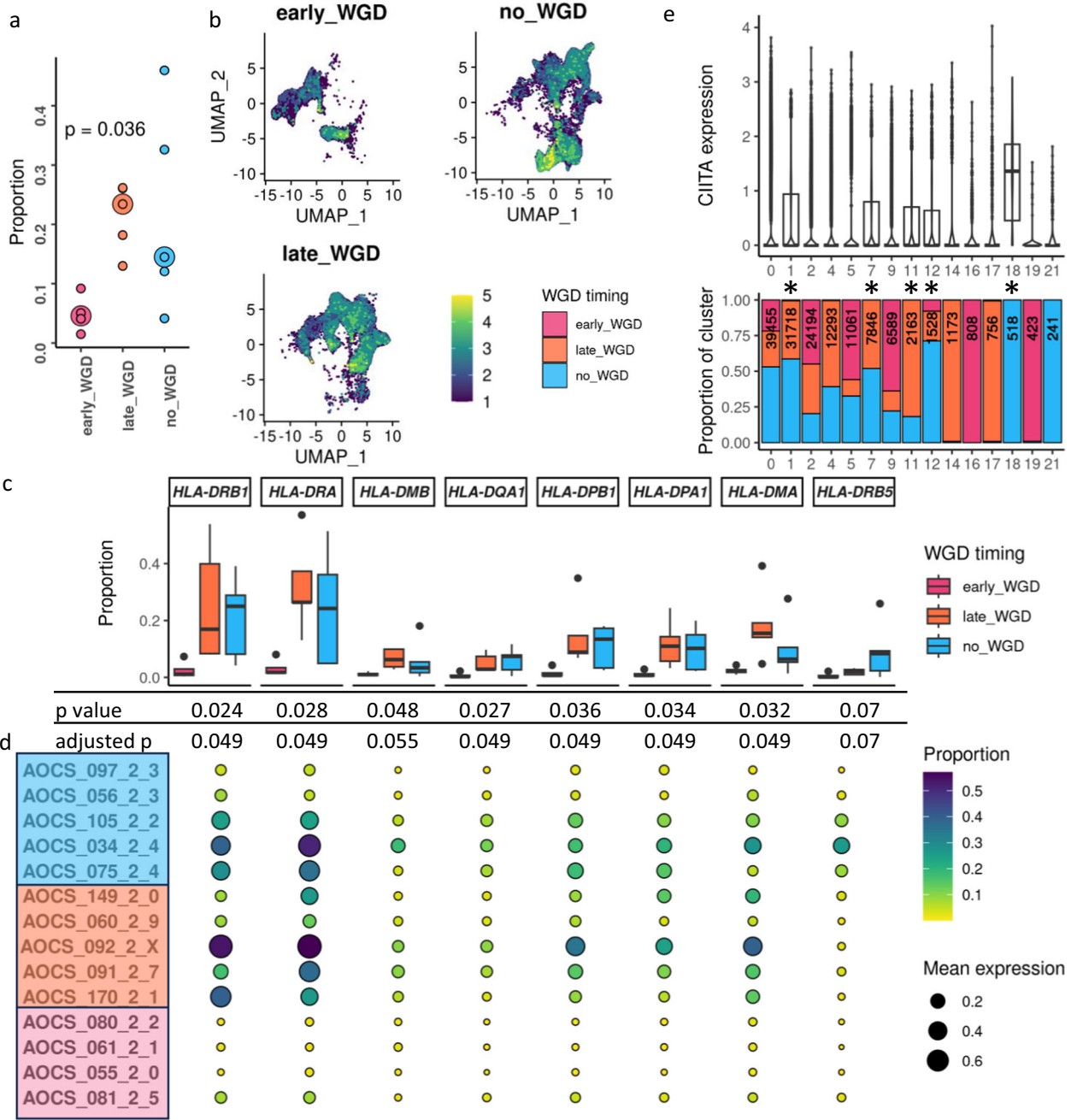

**Fig. 4 | *CIITA* and MHC-II gene expression in snRNAseq. a** Proportion of cancer cells expressing *CIITA*. Each small dot represents an individual patient. Large dot represents median for that category. Early WGD *n* = 4 patients; Late WGD *n* = 5 patients; No WGD *n* = 5 patients. Large dot represents median for that category. *p* = 0.036, Kruskal–Wallis test. **b** Feature UMAP plots of *CIITA* expression within cancer cells only, split by whole genome duplication (WGD) status. Cells are visualised in each plot from lowest to highest expression and numerical expression depicted as a relative scale to enable visualisation. **c** Boxplots summarising the proportion of cancer cells per patient expressing each MHC-II gene that was significant in the bulk DGE analysis. *P* values (Kruskal–Wallis test) and adjusted *p* values are depicted below. Early WGD *n* = 4 patients, 31,479 nuclei; Late WGD *n* = 5

patients, 48,685 nuclei; No WGD *n* = 5 patients, 60,602 nuclei. **d** Proportion of cancer cells expressing each gene shown for each patient individually, grouped by WGD timing. Mean expression denoted by size of circle (see legend). **e** Violin plots overlaying boxplots of *CIITA* expression in cancer cells by original Seurat cluster (top), annotated by proportion of cells from each WGD status category (bottom). Asterisks denote clusters designated to have substantial expression. For all boxplots: Left and right whiskers terminate at the minimum and maximum values no further than 1.5× interquartile range; centre line represents median (50th percentile); left and right boundaries of box represent the first (25th percentile) and third (75th percentile) quartiles, respectively; outlying values are plotted as individual points beyond whiskers. Source data are provided as a Source Data file.

was not uniform; interrogating the individual clusters revealed that *CIITA* was substantially expressed in only 5 of the 15 cancer cell clusters (Fig. 4e). These 5 clusters were strongly enriched for cells from tumours with late or no WGD (*p* = <0.001, Chi squared test), with only 1.0% of cancer cells from patients with early WGD falling into these 5 clusters, compared to 38.6% and 40.7% of late and no

WGD cancer cells respectively. Having seen a moderate correlation between *CIITA* and *IRF1* in the bulk RNAseq data, we examined this in the snRNAseq, but found a more modest correlation between *CIITA* and *IRF1* expression within cancer cells (R = 0.26, *p* < 0.001, CS-CORE, Supplementary Fig. 5a). All MHC-II genes which were statistically significant in the bulk RNAseq DGE analysis also had a

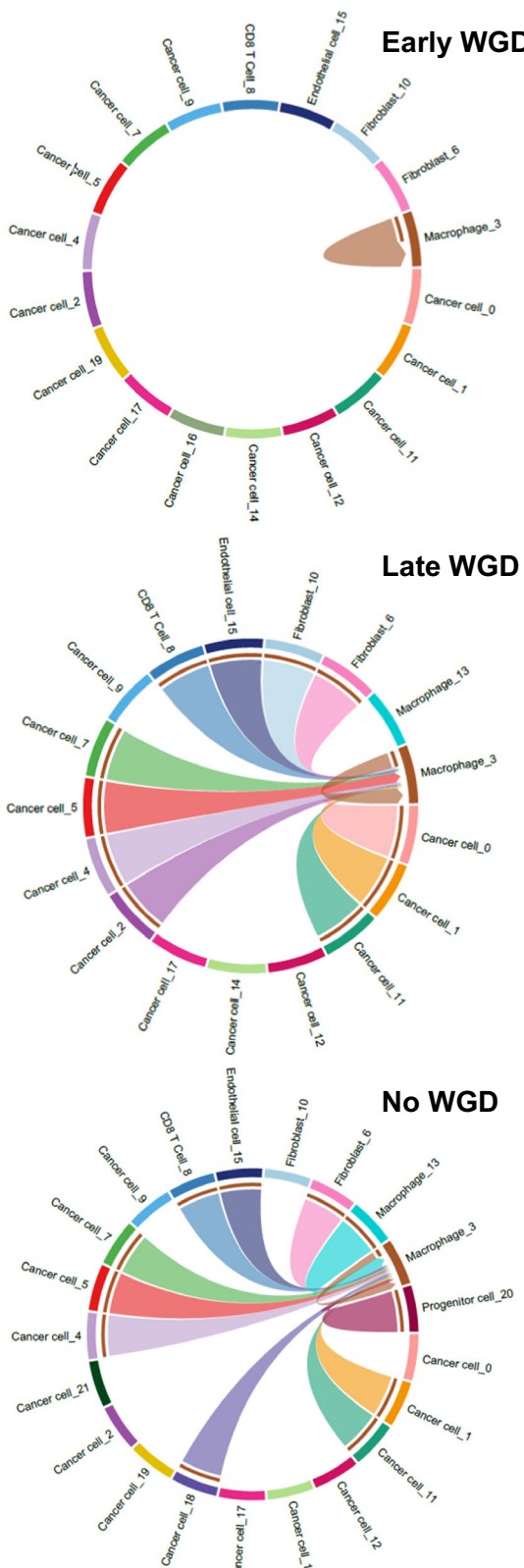

**Fig. 5 | Ligand-receptor pair communication predictions between cell clusters.**
Circle plots visualising the presence or absence of statistically significant interactions between and within cell groups for MHC-II (KEGG pathway hsa04514). The whole genome duplication (WGD) subgroups (early, late and no WGD) are individually depicted and not directly compared. Source data are provided in Supplementary Table 10.

statistically significant correlation with *CIITA* expression within cancer cells in the snRNAseq data (Supplementary Fig. 5a).

**The tumour microenvironment and its interactions with cancer cells**

In the snRNAseq data immune cells were observed in lower proportions in tumours with early WGD compared to those with late or no WGD, but this did not reach statistical significance ($p = 0.07$, Kruskal–Wallis test, Supplementary Fig. 5b, Supplementary Table 9).

We conducted a non-comparative examination of intercellular communications, identifying expression of statistically significant ligand-receptor pairs in the snRNAseq data between the cell clusters identified by Seurat using CellChat[43]; Supplementary Fig. 5c, Supplementary Table 10). Similarities across the three WGD categories were noted; for example, cancer-associated fibroblasts (Fibroblast_6) were the source of strong outgoing communications with both immune and cancer cells across all 3 WGD groups (Supplementary Fig. 5d), suggesting that they are a substantial contributor to modulation of the tumour microenvironment. Cancer cell_0, the largest cluster of cancer cells (28%), had few outgoing signals, little inter-cluster communication, and multiple but weak incoming signals. Conversely, many other cancer cell clusters such as Cancer cell_1 demonstrated multiple incoming and outgoing communications. These differences suggest that the individual cancer cell clusters represent unique populations of cancer cells, whereby some are relatively autonomous and less affected by the surrounding environment, whereas others are strongly modulated by communication with malignant and non-malignant cells.

In a descriptive analysis of interactions between cell subsets in the different WGD categories, tumours with early WGD had markedly low MHC-II signalling (Fig. 5). Only macrophages displayed any MHC-II signalling in tumours with early WGD; in contrast, extensive intercellular MHC-II signalling between macrophages and both cancer and non-malignant cells was observed in tumours with late or no WGD. While this is a non-comparative observation, this was also reflected in the per-gene pattern of MHC-II gene expression between WGD categories in cancer cells (Fig. 4c, d).

## Discussion

WGD is a frequent event in HGSC, yet its impact on tumour evolution is not well understood. Our findings suggest that early acquisition of WGD sculpts the molecular landscape of the tumour, resulting in decreased MHC-II expression compared to cancer cells with late or no WGD. Separately, we identified an association between the timing of WGD and patient survival outcomes. Together, these findings indicate a possible relationship between timing of WGD, MHC-II expression and clinical outcome in HGSC. While prior work has suggested there are worse outcomes in HGSC and other tumour types with WGD compared to those without, including more advanced stage at diagnosis and survival[1,5,44], the clinical relevance of WGD timing in HGSC has not previously been examined. Although we have not demonstrated a causative link between MHC-II expression and survival, it is plausible that they are directly related, given the strong association of immune cell infiltration and survival in HGSC, which has been recognised for many years[14]. We also identified cancer cell heterogeneity, with only a subset of cancer cells expressing MHC-II, likely driven by inducible *CIITA* expression, and that these cells are rare in cancers with early WGD.

Neoantigen expression is required to generate anti-tumour CD8 and CD4 T cell responses. Though loss of MHC-class I expression has been more strongly linked to cancer as a mechanism of immune escape[17,19], our findings demonstrate compelling evidence of tumour-specific MHC-class II dependent activity in HGSC and a potential pathway to immune escape in HGSC with early WGD. As cancer cells with WGD are genomically unstable and generally have a higher mutation burden than those without WGD[45,46], WGD is likely to result in

more immunogenicity due to a higher neoantigen burden and reduced MHC-II expression may be especially advantageous for survival of cancer cells with WGD. This is supported by observations of a relationship between aneuploidy or WGD and correlates of immune evasion in patient cohorts[3,23]. Furthermore, Davoli et al. described lower expression of pathways related to interferon gamma and immune regulation in tumours with high somatic copy number alterations across tumour types by GSEA analysis, including significantly lower *CIITA* expression[23]. Importantly they identified a relationship between high levels of aneuploidy and poor response to CTLA4 inhibition in patients with melanoma.

We did not identify a definite determinant of *CIITA* transcriptional control in this study. This is a topic which has been studied extensively in other tumour types[26,30,32,47]. It remains quite possible that hypermethylation of the pIV promoter of *CIITA* controls tumour-specific MHC-II expression, however we could not demonstrate this here, due to the limitations of genome-wide methylation arrays used in the included studies. Other limitations of this study include the inability to accurately evaluate tumours for their MHC-II specific neoantigen expression. In contrast to MHC-I, current methods of bioinformatically assessing neoantigens are suboptimal for MHC-II, in part due to the promiscuity of MHC-II molecules[48]. Additionally, it is unclear whether early acquisition of WGD occurs via a different mechanism to WGD acquired later and that this might explain the differences in patient survival and expression of *CIITA* and MHC class II genes between the groups. Alternatively, early WGD may simply reflect more time for the evolution of immune escape between WGD and the timing of sampling. Our findings illuminate an important gap in our knowledge of the immune response in HGSC.

Since the presence of tumour-specific MHC-II-expressing cancer cells is a predictor of ICI response in melanoma and triple negative breast cancer[20,22,49], our findings may have implications for therapeutic intervention in HGSC. Modest responses to ICI have been observed to date in ovarian cancer, but this may be due to the unselected nature of the ICI trials, and that they were used in later lines of treatment, by which time tumours have often evolved to become immune depleted[5,44]. The possibility of successful immune targeting agents with more MHC-II ligand specificity, such as LAG3 or TIM3, may be more fruitful in HGSC[27,50,51]. Since inducible *CIITA* expression is controlled epigenetically, hypomethylating agents may be able to reverse hypermethylation and abrogate the reduced *CIITA* and MHC-II expression in early WGD HGSC tumours[40,52]. It is clear from our previous work[6] that targeting resistance mechanisms earlier in the clinical course is vital to stem the development of a heterogenous cancer landscape with numerous resistance mechanisms that cannot all be plausibly targeted. Identifying early features of HGSC which are therapeutically targetable such as this is therefore clearly attractive and warrants further investigation.

## Methods
This research study was approved by the Peter MacCallum Cancer Centre Human Research Ethics Committee (15/84).

### Patient cohorts
**ICGC discovery cohort.** Copy number from WGS, bulk RNA sequencing (RNAseq), methylation arrays and clinical data from 79 primary HGSC samples from the ICGC dataset, which are described in Patch et al.[24], were utilised.

Single nuclei RNAseq was performed on 15 of the snap frozen primary tumour samples that were part of the ICGC study, described further below. Patients had previously given written informed consent through the Australian Ovarian Cancer Study (AOCS), approved by the Peter MacCallum Cancer Centre HREC (01/60). The samples were

chosen based on the WGD timing determined from the WGS, the availability of tissue and tumour cellularity. All participants in this study were female, and participants in AOCS were not provided with compensation.

**TCGA validation cohort.** TCGA data used in this paper, generated by the TCGA Research Network[53], was downloaded using the R package TCGAbiolinks (RRID:SCR_017683) v2.20.0 for RNA and copy number data, and v2.31.2 for Illumina Infinium HumanMethylation27 array data. ABSOLUTE purity estimates were downloaded directly from supplementary information from Aran et al., and copy number signatures from Steele et al.[54,55].

### Bulk RNAseq analysis
HTseq values from both cohorts were processed using edgeR (RRID:SCR_012802) v3.34.0 to generate trimmed mean of M values (TMM)[56,57]. Limma::voom (RRID:SCR_010943) v3.50.3[58] was used to generate normalised and log-transformed values for input into differential gene expression.

### Copy number and whole genome duplication analysis
Tumour purity and copy number log ratio were inferred from the ICGC WGS data by FACETS[25]. Whole genome duplication (WGD), defined by percentage of the autosomal genome with a major copy number of 2 or more, was calculated as per the method described by Bielski et al.[1]. The median tumour purity was 62% and 68.7% in those with and without WGD respectively, however we noted that cases excluded by our purity threshold of 0.3 were preferentially those without WGD. It is possible that this may therefore result in a proportionally higher frequency of WGD in our cohort, but does not bias the classification or analysis itself. Both copy number and RNAseq data had been aligned to reference genome GRCh37 for ICGC. TCGA harmonised GRCh38 data was downloaded as using TCGAbiolinks v2.20.0. and gene expression annotated to GRCh38. Timing of WGD was inferred using the method described by Dewhurst et al.[36].

### Differential gene expression analysis
**Covariates.** Gene expression can be affected by a number of technical factors, which should reasonably be considered in RNAseq analysis[3]. Technical factors, library preparation batch and sequencing run were considered as potential effects however these contributed little to bias and were not included. Accounting for copy number was particularly important, because, by definition, the genome-doubled samples were expected to have on average higher copy number and therefore gene expression, and this was done using the copy number log ratio.

Homologous recombination (HR) status can be quantified in a number of ways: HR gene annotation (*BRCA1*/2 annotation of somatic and germline variants), COSMIC SBS signature 3, HRDsum, Classifier of Homologous Recombination Deficiency (CHORD) and copy number signature 17 (associated with HRD)[55,59–61]. Having tested models encompassing each of these classifiers on a subset of genes spanning all chromosomes for efficient comparison, all models were found to fit better with a HR covariate than without, and there was little difference between Akaike Information Criterion (AIC) and F statistics between models. The model was therefore run with copy number signature 17, as the HR-deficiency covariate as complete information was available for both datasets.

**Model.** In order to account for copy number in the DGE analysis, which is a per-gene and per-sample value, standard tools such as edgeR or limma::voom could not be used for DGE analysis. Therefore, a GLM was used, using HR status, copy number at that gene segment and tumour

purity as covariates, with a gaussian distribution, using the R packages foreach v1.5.2 and doParallel v1.0.17[62] in the format:

$$\text{Normalised log-transformed voom value} \sim \text{WGD} + \text{HR covariate} + \text{purity} + \text{copy number log ratio}$$

*P* values were adjusted for multiple hypothesis testing using the Benjami−Hochberg method. The coefficient estimate was used as the measure of magnitude of effect.

Because the models were run iteratively per gene, rather than as one, diagnostics could not be applied to the entire model. As an alternative, three genes were randomly picked (genes falling at tertiles) and their individual models assessed using DHARMa[63]. Model fit was assessed using DHARMa (RRID:SCR_022136) v0.4.5, specifically assessing for Kolmogorov-Smirnov goodness-of-fit test, outliers and dispersion.

**Pathway analysis.** ActivePathways v1.1.1, which detects over-representation of genes using a ranked hypergeometric test, was used for pathway analysis, due to its advantage of being able to integrate a discovery and validation dataset[35]. A coefficient estimate of >±1 was used for the ICGC dataset, but a less stringent threshold of ±0.5 for the TCGA dataset, to enable inclusion of a sufficient number of genes.

**CIBERSORTx.** The CIBERSORTx web-based pipeline (RRID:SCR_016955) was previously used to infer immune cell estimations from RNA sequencing, described in Burdett et al.[6,64]. Briefly, this was performed with batch correction enabled, B-mode, quantile normalisation was disabled, absolute mode and 500 permutations, and the LM22 signature matrix was employed for immune cell deconvolution. No further processing was performed.

### Methylation data
Methylation array data was generated in our previous work[6,24]. This included all 79 cases described here and no further processing was done[6]. Probes corresponding to promoter regions were identified as per our previous work, assessing for probes within 2000bp of the transcription start site and an inverse spearman correlation of ≤−0.3. Beta values of ≥0.8 were considered to be hypermethylated.

### Single nuclei RNA sequencing
**RNA pre-processing.** Single nuclei were isolated from snap frozen tumour samples using the protocol established by Martelotto et al.[65], with minor modification as summarised: snap frozen samples were sectioned at 100 μm and chilled salty EZ lysis buffer (supplemented with RNase inhibitor) was added. The samples were homogenised using a douncer then filtered using a 70 μm strainer mesh, to remove the undigested and fatty tissues. The nuclei were then centrifuged at $500 \times g$ for 5 min at 4 °C. To clean up the nuclei suspension, gradient solution was used and centrifuged $3200 \times g$, for 20 min at 4 °C. The nuclei were washed in Wash Buffer 2 (10 mM Tris-HCl pH 7.5, 10 mM NaCl, 3 mM MgCl$_2$, 1% BSA, 0.2–1 U/uL Protector RNAse Inhibitor (Roche; cat# 3335399001) and passed through 40 μm strainer mesh. The nuclei were resuspended in Wash Buffer 1 (1× PBS, 1% BSA, 0.2–1 U/μL Protector RNAse Inhibitor).

The 10× Chromium platform was used to generate single nuclei RNAseq (snRNAseq) libraries using the 10x Genomics single cells 3′ reagent kit v3.1 (cat# 1000123). Libraries were sequenced on the Illumina NovaSeq6000 (paired end 150 bp reads, 40,000 reads per cell).

**Data and processing.** snRNAseq data for 14 samples was aligned to the reference genome GRCh38 using CellRanger (RRID:SCR_023221)[66]. Seurat (RRID:SCR_016341) v4.1.1 was used to process, scale and normalise the data, with default filters (min.cells = 3, min.features = 200). More advanced filtering removed low quality cells, multiplets, dead cells and those with a high mitochondrial RNA content (filtered to <15%).

Further transformation and downstream analyses were performed using Seurat v4.3.0.1. Threshold based filtering was applied, to remove the most extreme (and improbable) values, so that Seurat nFeature and nCount parameters more than ±2.5 standard deviations from the mean were excluded.

Data was then processed using SCTransform on individual samples, scaled and then batch-corrected using Harmony (RRID:SCR_022206) v0.1.0[67,68]. Correction factors used were processing batch and sequencing run. Seurat was also used to run UMAP, with 25 dimensions.

**Cell annotation, copy number variation and cell-cell communication.** Choice of cell annotation methods may have a strong impact on results and subsequent conclusions. A consensus method was therefore used: first, ScType was used with two separate datasets chosen for their respective strengths. The internally curated set of immune markers within ScType performed well to annotate non-cancer cells, and the markers relevant to an ovarian cancer context identified by[69]. performed well to identify cancer cells and provide further weight to in-built ScType annotations. Independently, expression of marker genes for both cancer and non-malignant cell types (such as *EPCAM, PAX8, MUC16, WT1* for cancer cells and *COL1A1* for fibroblasts) were examined per cluster to assess the fidelity of these calls. A consensus based on the 3 annotations was then taken.

Cells were further classified using predictions of whether the cells were diploid or aneuploid using an integrative Bayesian segmentation approach via CopyKAT (RRID:SCR_024512)[70]. Specifically, cell clusters annotated as cancer or non-malignant cells were assessed for the proportion of cells predicted to be aneuploid or diploid, with cells classified as 'not defined' excluded. Cancer cells were then reprocessed, using the NormalizeData function with Seurat, since we found that SCTransform is valuable for cell type inference but yields data which has a continuous yet binned structure which is difficult for downstream statistical analyses on gene expression. Data was then batch-corrected for processing and sequencing batch with Harmony and UMAP rerun with 18 dimensions.

CellChat (RRID:SCR_021946) was used to analyse ligand-receptor pairs and predict incoming and outgoing signals between cell subsets[43].

### Statistics
All analysis and statistics were performed in RStudio (RRID:SCR_000432, v4.1.0). The type of numeric statistical tests used are indicated in text. Analyses were run with ggpubr (RRID:SCR_021139, v0.4.0) for data which did not include multiple samples/cells from the same patient. All comparisons of means are two-sided. For statistical analysis of gene expression in cancer cells in snRNAseq data, a generalised linear mixed model with the 'tweedie' distribution was used[71,72].

Mixed models were conducted using glmmTMB (v1.1.3) and fit assessed with DHARMa (v0.4.5). Categorical tests were run using Chi square test (stats v4.1.0). To address the issue of sparse matrices, correlations on snRNAseq were performed with CS-CORE[73] (v1.0.1).

The threshold for statistical significance across all analyses was $p < 0.05$. For CellChat, the default number of permutations (M = 100) was used, with a $p$ value of <0.05 considered significant. CellChat internally uses bonferonni correction for multiple testing correction for differential interaction analysis. The final p values are not corrected again after the permutation as per the authors' documentation.

All other multiple testing corrections were performed using Benjamini−Hochberg method. Joint $p$ values from glmms were calculated using emmeans package (v1.7.5).

**Table 1 | Immunohistochemistry scoring**

| Parameter | Options |
|---|---|
| Cell type stained | Stromal lymphocytes (L)<br>Tumour infiltrating lymphocytes (TIL)<br>Tumour cells (T) |
| Intensity | Positive: 1–3+ |
| Percentage of assessed cells staining | Numerical percentage |

## Immunohistochemistry

Immunohistochemistry (IHC) was performed using the anti-HLA DR + DP + DQ antibody [clone CR3/43] (Abcam Cat# ab7856 lot# GR3434335-1; RRID:AB_306142). Following optimization on ovarian tumour tissue and non-malignant tonsil sections, the IHC was performed on 2 tissue microarrays (TMAs) comprising two cores per primary tumour from 82 patients from the ICGC study.

TMAs were incubated at 56 °C for 30 min to deparaffinize them. Slides were dewaxed in histolene, then passed through 100% ethanol, and then passed through 70% ethanol for 1 min, and finally deionized water for 5 min. Slides were transferred to a pressure cooker to undergo heat-mediated antigen retrieval, by heating to 125 °C for 3 min in sodium citrate buffer (pH 6.0). Slides were then washed in deionized water. Endogenous peroxidase activity was quenched by submerging slides in 3% hydrogen peroxide and deionized water. Slides were rinsed in Tris-buffered saline with 0.1% Tween 20 detergent (TBST), then blocked using 1X Antibody Diluent/Block (Akoya Biosciences, ARD1001) for 5 min at room temperature. The primary antibody was used at a dilution of 1:150. 100 μL of the prepared dilution was added to each slide and incubated for 60 min at room temperature. Slides were then washed three times for 2 min in TBST. Slides were then incubated for 30 min with the secondary antibody (Anti-Mouse Immunoglobulin Anti-HLA, Vector Laboratories, catalogue number MP-7401-15), then rinsed three times for 2 min with TBST at room temperature. Next, slides were incubated in chromagen substrate solution (25 μL/1 drop 3,3'-Diaminobenzidine (DAB) (Agilent, K3467) with 1 mL substrate K3468), until desired stain intensity developed, at <1 min. Slides were rinsed in deionized water and counterstained with haematoxylin.

Reporting of slides was performed by an external pathologist trained in gynaecological oncology, with scoring parameters summarised in Table 1.

Each tumour sample was represented on a TMA in duplicate; after staining only 45 of the cases had 2 cores (one had been sampled twice resulting in 46 pairs, with a total of 109 cores).

## Reporting summary

Further information on research design is available in the Nature Portfolio Reporting Summary linked to this article.

## Data availability

All bulk RNAseq, whole genome sequencing and methylation data used in this study has previously been published. The ICGC publicly available data[24,74] used in this study are available in the European Genome-Phenome archive under accession code EGAD00001000877. The TCGA publicly available data[53] used in this study are available from https://portal.gdc.cancer.gov/. Due to the sensitive nature of these patient datasets, access is subject to approval from the ICGC Data Access Compliance Office (https://docs.icgc.org/download/data-access/). ICGC methylation data sets have been deposited into the Gene Expression Omnibus (GEO; https://www.ncbi.nlm.nih.gov/geo/) under accession code GSE65821, without access restrictions. ICGC gene count level transcriptomic data has been deposited into the GEO under accession code GSE209964. ABSOLUTE purity estimates and copy number signatures were downloaded directly from supplementary information[54,55]. The snRNAseq raw data generated in this study

have been deposited in the European Genome-Phenome archive under the accession code EGAD50000000364. Due to the sensitive nature of these patient datasets, the data is available under restricted access, which can be obtained by contacting DGO@petermac.org. Responses to data requests aim to be provided within two weeks. Access will be granted for appropriate research use, which are in line with the original consent provided through AOCS. Duration of data access once granted is not restricted. Processed data are available in Supplementary Tables. The remaining data are available within the Article, Supplementary Information or Source Data file. Source data are provided with this paper.

## Code availability

This study uses existing R packages for all analyses and no novel/custom code was developed. The code to generate figures and GLM has been uploaded to synapse.org at https://www.synapse.org/Synapse:syn52673607 (free Synapse account is required) and is available on request to corresponding author.

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

## Acknowledgements

We acknowledge the statistical input of Madawa Jayawardana with regard to complex model fitting. We thank Peter MacCallum Cancer Centre Molecular Genomics Core facility, supported by the Australian Cancer Research Foundation. The Australian Ovarian Cancer Study (AOCS) acknowledges the cooperation of the participating institutions in Australia and acknowledges the contribution of the study nurses, research assistants and all clinical and scientific collaborators to the study. The complete AOCS Study Group can be found at www.aocstudy.org. We would like to thank all of the women who participated in these research programs. This research was financially supported by grants from Department of Health and Human Services through the National Health and Medical Research Council of Australia (NHMRC, APP1189939 to N.L.B., APP1161198 to D.D.L.B. and E.L.C.) and the Victorian Cancer Agency (VCA MCRF21004 to E.L.C.). The AOCS Group was supported by the U.S. Army Medical Research and Materiel Command under DAMD17-01-1-0729, The Cancer Council Victoria, Queensland Cancer Fund, The Cancer Council New South Wales, The Cancer Council South Australia, The Cancer Council Tasmania and The Cancer Foundation of Western Australia (Multi-State Applications 191, 211 and 182) and NHMRC (ID199600; ID400413 and ID400281). The Australian Ovarian Cancer Study gratefully acknowledges additional support from Ovarian Cancer Australia and the Peter MacCallum Cancer Foundation. Figure 1a was created with BioRender.com released under a Creative Commons Attribution-NonCommercial-NoDerivs 4.0 International license.

## Author contributions

Using CRediT (Contribution Roles Taxonomy) N.L.B.—conceptualization, formal analysis, investigation, software, visualization, writing original draft, writing—review/editing M.O.W.—investigation, writing—review/editing A.P.—software, formal analysis, writing—review/editing L.T.—software, formal analysis, writing—review/editing S.A.—investigation, writing—review/editing AOCS—data curation, resources, writing—review/editing D.D.L.B.—conceptualization, resources, funding acquisition, writing—review/editing E.L.C.—conceptualization, funding acquisition, supervision, writing original draft, writing—review/editing.

## Competing interests

D.D.L.B. reports research support grants from Roche-Genentech, AstraZeneca, and personal consulting fees from Exo Therapeutics, none of which are related to this work. E.L.C reports research grant funding from AstraZeneca which is not related to this work. The remaining authors declare no potential conflicts of interest.

## Additional information

## Australian Ovarian Cancer Study Group

**Management Group** D. Bowtell[1,2], G. Chenevix-Trench[5], A. Green[5], P. Webb[5], A. DeFazio[6,7,8] & D. Gertig[9]

**Project and Data Managers** N. Traficante[1,2], S. Fereday[1,2], S. Moore[5], J. Hung[6], K. Harrap[5], T. Sadkowsky[5] & N. Pandeya[5]

**Research Nurses and Assistants** L. Bowes[1], L. Galletta[1], D. Giles[1], J. Hendley[1], K. Alsop[1,2], B. Alexander[5], P. Ashover[5], S. Brown[5], T. Corrish[5], L. Green[5], L. Jackman[5], K. Ferguson[5], K. Martin[5], A. Martyn[5], B. Ranieri[5], M. Malt[5], Y. E. Chiew[6], A. Stenlake[8], H. Sullivan[8], A. Mellon[10], R. Robertson[10], T. Vanden Bergh[11], M. Jones[11], P. Mackenzie[11], J. Maidens[12], K. Nattress[13], J. White[14], V. Jayde[15], P. Mamers[16], T. Schmidt[17], H. Shirley[17], S. Viduka[17], H. Tran[17], S. Bilic[17], L. Glavinas[17], C. Ball[18], C. Young[18] & J. Brooks[19]

**Clinical and Scientific Collaborators** L. Mileshkin[1,2], G. Au-Yeung[1,2], K. Phillips[1,2], D. Rischin[1,2], N. Burdett[1,2,3], R. Delahunty[1], E. Christie[1,2], D. Garsed[1,2], S. Fox[1], D. Johnson[1], S. Lade[1], M. Loughrey[1], N. O'Callaghan[1], W. Murray[1], D. Purdie[5], D. Whiteman[5], A. Proietto[10], S. Braye[10], G. Otton[10], C. Camaris[11], R. Crouch[11], L. Edwards[11], N. Hacker[11], D. Marsden[11], G. Robertson[11], D. Bell[12], S. Baron-Hay[12], A. Ferrier[12,50], G. Gard[12], D. Nevell[12], N. Pavlakis[12], S. Valmadre[12], B. Young[12], P. Beale[13], J. Beith[13], J. Carter[13], C. Dalrymple[13], R. Houghton[13], P. Russell[13], M. Davy[14], M. K. Oehler[14], C. Hall[14], T. Dodd[14], P. Blomfield[15], D. Challis[15], R. McIntosh[15], A. Parker[15], D. Healy[16], T. Jobling[16], T. Manolitsas[16], J. McNealage[16], P. Rogers[16], B. Susil[16], E. Sumithran[16], I. Simpson[16], N. Zeps[17], I. Hammond[18], Y. Leung[18], A. McCartney[18,51], R. Stuart-Harris[20], F. Kirsten[21], J. Rutovitz[22], P. Clingan[23], J. Shannon[24], T. Bonaventura[25], J. Stewart[25], S. Begbie[26], A. Glasgow[26], M. Friedlander[27], M. Links[28], J. Grygiel[29], J. Hill[30], A. Brand[7,31], K. Byth[31], P. Harnett[7,31], G. Wain[31], R. Jaworski[32], R. Sharma[7,32], B. Ward[33], D. Papadimos[33], A. Crandon[34], M. Cummings[34], K. Horwood[34], A. Obermair[34], L. Perrin[34], D. Wyld[34], J. Nicklin[34,35], T. Healy[36], K. Pittman[36], D. Henderson[37], J. Miller[38], J. Pierdes[38], B. Brown[39], R. Rome[39], D. Allen[40], P. Grant[40], S. Hyde[40], R. Laurie[40], M. Robbie[40], P. Waring[41], V. Billson[42], J. Pyman[42], D. Neesham[42], M. Quinn[42], C. Underhill[43], R. Bell[44], L. F. Ng[45], R. Blum[46], V. Ganju[47], M. Buck[48] & I. Haviv[49]

[5]QIMR Berghofer Medical Research Institute, Brisbane, QLD 4006, Australia. [6]Centre for Cancer Research, The Westmead Institute for Medical Research, Sydney, NSW 2145, Australia. [7]The University of Sydney, Sydney, NSW 2006, Australia. [8]Department of Gynaecological Oncology, Westmead Hospital, Sydney, NSW 2145, Australia. [9]Melbourne School of Population and Global Health, University of Melbourne, Parkville, VIC 3052, Australia. [10]John Hunter Hospital, Lookout Road, New Lambton, NSW 2305, Australia. [11]Royal Hospital for Women, Barker Street, Randwick, NSW 2031, Australia. [12]Royal North Shore Hospital, Reserve Road, St Leonards, NSW 2065, Australia. [13]Royal Prince Alfred Hospital, Missenden Road, Camperdown, NSW 2050, Australia. [14]Royal Adelaide Hospital, North Terrace, Adelaide, SA 5000, Australia. [15]Royal Hobart Hospital, 48 Liverpool St, Hobart, TAS 7000, Australia. [16]Monash Medical Centre, 246 Clayton Rd, Clayton, VIC 3168, Australia. [17]Western Australian Research Tissue Network (WARTN), St John of God Pathology, 23 Walters Drive, Osborne Park, WA 6017, Australia. [18]Women and Infant's Research Foundation, King Edward Memorial Hospital, 374 Bagot Road, Subiaco, WA 6008, Australia. [19]St John of God Hospital, 12 Salvado Rd, Subiaco, WA 6008, Australia. [20]Canberra Hospital, Yamba Drive, Garran, Canberra, ACT 2605, Australia. [21]Bankstown Cancer Centre, Bankstown Hospital, 70 Eldridge Road, Bankstown, NSW 2200, Australia. [22]Northern Haematology & Oncology Group, Integrated Cancer Centre, 185 Fox Valley Road, Wahroonga, NSW 2076, Australia. [23]Illawarra Shoalhaven Local Health District, Wollongong Hospital, Level 4 Lawson House, Wollongong, NSW 2500, Australia. [24]Nepean Hospital, Derby Street, Kingswood, NSW 2747, Australia. [25]Newcastle Mater Misericordiae Hospital, Edith Street, Waratah, NSW 2298, Australia. [26]Port Macquarie Base Hospital, Wrights Road, Port Macquarie, NSW 2444, Australia. [27]Prince of Wales Clinical School, University of New South Wales, NSW 2031, Australia. [28]St George Hospital, Gray Street, Kogarah, NSW 2217, Australia. [29]St Vincent's Hospital, 390 Victoria Street, Darlinghurst, NSW 2010, Australia. [30]Wagga Wagga Base Hospital, Docker St, Wagga Wagga, NSW 2650, Australia. [31]Crown Princess Mary Cancer Centre, Westmead Hospital, Westmead, Sydney, NSW 2145, Australia. [32]Department of Pathology, Westmead Clinical School, Westmead Hospital, The University of Sydney, NSW 2006, Australia. [33]Mater Misericordiae Hospital, Raymond Terrace, South Brisbane, QLD 4101, Australia. [34]The Royal Brisbane and Women's Hospital, Butterfield Street, Herston, QLD 4006, Australia. [35]Wesley Hospital, 451 Coronation Drive, Auchenflower, QLD 4066, Australia. [36]Burnside Hospital, 120 Kensington Road, Toorak Gardens, SA 5065, Australia. [37]Flinders Medical Centre, Flinders Drive, Bedford Park, SA 5042, Australia. [38]Queen Elizabeth Hospital, 28 Woodville Road, Woodville South, SA 5011, Australia. [39]Freemasons Hospital, 20 Victoria Parade, East Melbourne, VIC 3002, Australia. [40]Mercy Hospital for Women, 163 Studley Road, Heidelberg, VIC 3084, Australia. [41]Department of Pathology, University of Melbourne, Parkville, VIC 3052, Australia. [42]The Royal Women's Hospital, Parkville, VIC 3052, Australia. [43]Border Medical Oncology, Wodonga, VIC 3690, Australia. [44]Andrew Love Cancer Centre, 70 Swanston Street, Geelong, VIC 3220, Australia. [45]Ballarat Base Hospital, Drummond Street North, Ballarat, VIC 3350, Australia. [46]Bendigo Health Care Group, 62 Lucan Street, Bendigo, VIC 3550, Australia. [47]Peninsula Health, 2 Hastings Road, Frankston, VIC 3199, Australia. [48]Mount Hospital, 150 Mounts Bay Road, Perth, WA 6000, Australia. [49]Faculty of Medicine, Bar-Ilan University, 8 Henrietta Szold St, Safed, Israel. [50]Deceased: A Ferrier. [51]Deceased: A McCartney.

