## [Peer Review File · Nature Communications]

Timing of whole genome duplication is associated with tumor-specific MHC-II depletion in serous ovarian cancerReviewers' Comments:

Reviewer #1:

Remarks to the Author:

Burdett et al present an interesting study and shed new light on potential vulnerabilities of high grade serous ovarian cancer (HGSC) tumours with whole genome duplication (WGD). Using bulk RNAseq data from ICGC and TCGA HGSC cohorts and customised differential gene expression analyses, they identify MHC-II related genes to be significantly downregulated/depleted in HGSC tumours with WGD compared to non-WGD tumours and associate this with early-onset WGD, suggesting that the timing of WGD is important and shapes the molecular landscape and thus the immune profile in HGSC. They further follow-up their findings with a subset of single-cell RNA sequencing of 14 tumours and report that reduced MHC-II expression is driven by subsets of cancer cells (rather than canonical antigen-presenting cells).

The manuscript is very well written, describes novel findings and will be of significant interest to the fields of ovarian cancer, cancer genomics and immuno-oncology, and I am therefore very supportive of this study being published in Nature Communications. However, there are some major and a few minor aspects, that would need to be addressed and/or could improve the quality of this manuscript prior to publication.

Major points:

1. Can the authors further clarify their thoughts and reasoning for removing gene specific copy number effects from the differential gene expression analysis and contrast their findings to results obtained to a similar model not taking gene-specific copy number log ratios into account (i.e. include genes that were upregulated as a result of e.g. focal amplifications)? Various studies have now shown that copy number events are not perfectly correlated with gene expression, and gene expression driven or altered by copy number alterations might still present and reveal valuable vulnerabilities associated with WGD. Would the authors find more upregulated genes or more upregulated hallmark pathways if copy number log ratio was removed as a covariate from the applied model? And would MHC-II related genes still be significantly down-regulated? At a minimum, please justify the reasoning for removing this copy number from this analysis in more (data-driven) detail.
2. Line 106-107 – Please include this data as plot/table in the manuscript, potentially as supplementary figure. (See other examples of this below. In my opinion all results/conclusions and statistics should be supported by data/figures in the manuscript).
3. Methylation data is also available from TCGA. Would it be possible to utilise this from the TCGA ovarian cancer cohort in addition to the methylation array data from references 6 and 19, to investigate in more detail the methylation status of the mentioned four main promoters? Is CIITA more frequently hypermethylated in WGD tumours in this dataset? Including this analysis might be very relevant and interesting, especially since this is further discussed as a potentially targetable vulnerability in the discussion (line 364-367)
4. Line 135 – Include figure or table to show correlation between IRF-1 and CIITA. It might also be worth to briefly clarify whether IRF1 is a positive or negative regulator of CIITA in the main text.
5. Line 190-197. The hypothesis that MHC-II expression might result in poorer PFS is not directly addressed in the presented analysis and figures. Please either re-word the hypothesis or re-do the analysis looking at MHC-II expression and PFS/OS instead. It might also be worth to perform a coxPH analysis taking multiple covariates into account, including WGD/HRD status, stage, age etc.
6. Line 206 – “data not shown”. Again, please include data and/or figures.
7. Line 212 – IHC data. Please specify how this was quantified? Could IHC staining be measured using

computational image analysis (such as the Halo software) to (potentially more robustly) quantify MHC-II staining/intensity? Please specify how (and from what measurements/quantifications) the p-values were derived. Please also include both of the above in the relevant Methods section, and figure legend. In addition, which figure is this statement and the p-values (in line 215) referring to? If this is referring to Figure 2c please include the figure reference here.

8. Line 222-223 – mentioning 15 patient samples to be sequenced but data for only 14 samples is shown. The following sentence also only mentions 14 patients. Did one fail? If so why? Correct numbers in main text and methods section, and/or clarify if and why one sample may have failed or was excluded from the analyses.

9. Line 244 and 246– Please include figure/data to support these observations?

10. Line 255-257 – Please show data/figure for this in supplement.

11. Line 296-299 – NRG, Collagen and Laminin seem to be the top 3-4 hits for tumours from all three WGD categories. Is the observation described here significant? Of note also that the scales of the grey row bars (x-axis) are significantly different (>10-fold) between the three plots. Could the authors please comment on this and its meaning in the main text and/or figure legend?

12. In the discussion, the authors acknowledge limitations in current methods for assessing MHC-II specific neoantigen expression. However, it might be worth to estimate total neoantigen load (number of neoantigen candidates identified using tools such as pVacSeq) and compare the neoantigen load to MHC-II expression and WGD status in HGSC. I believe the TCGA ovarian cancer cohort should provide sufficient data for this analysis, and the results would be very interesting in the context of the main manuscript findings and the discussion points in lines 327-335.

Minor points:

1. Line 56 “and therefore survival for almost 20 years” – could the authors reword this sentence to avoid confusion. What do the 20 years refer to, the survival or evidence?

2. The use of the testing (ICGC) and validation (TCGA) datasets seem slightly inconsistent throughout the first part of the manuscript (results shown in Fig. 1 – Fig. 2a). Could the authors sign-post and justify in the text more clearly which dataset was used when and why (see some additional individual the minor comments) and potentially include results from both datasets when only one is shown in supplementary figure?

More specifically, Fig 1a only shows ICGC samples (could include TCGA samples here as well); Fig 1b shows both datasets separately; Fig 1c shows both combined/integrated data; Fig 2a shows ICGC only; Fig 2b again shows both combined/integrated data). For example, could the authors justify the reasoning for integrating the two datasets in line 138-141. I am assuming that this is due to statistical power and sample numbers, but this should still be mentioned in the main text/figure legend.

3. Line 159-173 – Include TCGA results two (either as joined or separate analysis) or alternatively justify why excluded. Compare and contrast results.

4. Line 145 – very briefly describe what the Quinton et al study is if mentioning it explicitly in the main text.

5. Line 242 – “even after accounting for patient-specific differences” – please specify what these differences were and how they were accounted for.

6. Figure 3c – lower panel. Could the y-axis be changed/transformed to sqrt of log₁₀p scale, and all data points be shown (instead of just outliers)? Or alternatively show violin plots so the distribution of

the data becomes visible.

7. Figure 3d – If possible please don't use the same colour for the overall cell count as is used for the no_WGD group since this might be confusing/misleading. Could the colour be changed to e.g. grey? Could the authors also explain the discrepancy of the number of cells shown in the top cell counts to the bottom cell counts? If there are cells that were unassigned, could this be included? Also, would it be possible to change the lower cell count (by cell type) to proportions (i.e. y-axis of 0 to 1) to make the difference in cell type abundance more clear and visible, especially since the cell count is already indicated in the top barplot?

Reviewer #2:

Remarks to the Author:

Burdett et al. build on their prior finding that the timing of WGD in HGSC is a driver of tumor heterogeneity and patient outcomes. They leverage previously published bulk sequencing data from multiple cohorts to show that early WGD is associated with lowered expression of MHC II genes and correlates with low expression of the CIITA transcription factor that is known to regulate MHC II expression. They also perform single nuclear RNA sequencing to show cancer cells are one of the key cell types in tumors that have early WGD to lose MHC II expression.

Major Concerns:

In general, statistics are lacking for many of the reported findings. Importantly, for the single cell data comparisons should be made at the per-patient level and corresponding statistical test results provided for each comparison. It does not appear that any type of correction for multiple testing has been performed and considering the very large number of statistical tests performed, some method of correction is needed. There is also no description in the methods showing clearly for each type of test what the significance threshold was.

Cancer cells should be reclustered and analyzed separately for Figure 3.

Minor Concerns:

The introduction could include more discussion on what is known about MHC II expression by cancer cells and the consequences for immune recognition and response to checkpoint blockade.

Page 2, line 44 – it's unclear just what is meant here. If the use of "continues to be acquired" refers to an increase over disease progression as a result of comparing primary tumors with end stage, then those analyses should be done, and the results presented. Otherwise this sentence should be modified or removed.

Page 2 lines 80-81. What is the significance threshold used to determine the number of DEGs? This is a very large number of DEGs for an experiment of this type. A full list of DEGs needs to be provided.

Page 3, lines 82-86. A presentation of the results of the purity analysis should be provided as tables and figures prior to the DEG analysis so that the potential of the covariates can be assessed by the reader.

Page 4, lines 100-112. Please split the cohort into 4 groups: WGD+ with loss of Chr 6, WGD- with no loss of chr 6, WGD- with loss of Chr 6, WGD- with no loss of Chr 6, and then show expression of these genes in each of these 4 groups. Visualizing these values would be very helpful.

Page 4 lines 116-117. 166 HGSC tumors is a small subset of TCGA – how were these tumors selected in?

Page 5, line 139. Please show the full results of pathway analysis in ICGC and TCGA separately, and then when combined.

Please show the comparison of the Dewhurst and previously published PCAWG WGD estimation so that the reader can assess if the two methods in fact had good concordance by indicating samples profiled with both methods.

Page 5, lines 165-173: Recent single cell data have shown the molecular subtypes previously reported (and this citation is not the originating paper for the molecular subtypes anyway), are largely driven by sample composition, and are reflective of stroma content in the tumor, which is where immune cells reside in HGSC.

Figure 2: Please also provide a heatmap of gene expression data showing the top 1000 variable genes across the samples.

Page 7 lines 189-192: This analysis should be restricted to the 116 TCGA tumors in analysis previously in this report.

Page 7, lines 210-217: Please include the complete set of CIBERSORT annotations. A heatmap with corresponding table showing cell types and quantifications across the samples should be included.

Show dot plot of CIITA, MHC expression by cluster in 3C

Boxplots showing proportion of total cells for each cell type should replace 3d. Statistical analysis should be applied to show proportions are different.

Please include why 15 samples were used for scRNA-Seq but data is only shown for 14 samples.

What is the cell level correlation of CIITA and MHC? Cluster comparisons are made but what about on the cell level. Does low CIITA correlate with low MHC II across tumor cells?

Figure 3: Given previous publications showed that tumor cell clusters in HGSC are typically from a single patient please provide Figure 3a with cells labelled with Patient ID as a main figure and not supplemental. A discussion of why this may be different in your report should be included in this manuscript. It appears as though there are significant numbers of non-tumor cells with WGD. Is this an artifact of 2D visualization? Can a third dimension be added to these plots to help clarify? Particularly for cells like CD8 T cells, endothelial cells and fibroblasts.

Page 10, lines 248-251. Please provide a UMAP showing the location of cancer cells expressing CIITA.

There are no statistics for Figure 4. If the authors are claiming cell chat predicted MHC signaling is greater in the no or late WGD cells then they need to show statistics comparing per patient levels of this signal.

Statistics need to be provided for both figure panels in 5 to show on a patient level the reported differences are significant.

Were the samples in this cohort of the same stage and grade?

There is no indication which of these samples are HRD vs HRP, and this is an important factor in the TME of HGSC. Please provide the HR status of each tumor.

There is no data availability statement.

Reviewer #4:

Remarks to the Author:

Burdett and coauthors reported that serous ovarian cancers that have undergone whole-genome duplication early during their development show reduced MHC-II expression. I find the analyses (including the addition of single-nucleus RNA-Seq) to be rigorous and thoughtful, and the biological conclusion to be of broad interest to the field. Therefore I recommend the paper for publication at Nature Communications without major changes. Below are my comments that the authors may want to consider for additional clarification in a revised study.

1. The authors discussed the caveats of performing differential gene expression analysis for tumors with large segmental copy number changes.

1a. In the general linear model shown in Fig. 1a and also in method, is the "Norm. HTseq value" on the left hand side the log-transformed read count? It does not make sense to me if this is the normalized count (without log transformation).

1b. What was being used in the copy number logR? I would take this to be the normalized DNA copy number, i.e., segmental copy number divided by ploidy. From line 83-86, the authors mentioned that "by definition tumors with WGD will have higher total copy number per gene." I do not think this matters as both gene expression (l.h.s. of the linear model) and DNA copy number (r.h.s. of the linear model) are relative measurements. As a simple example, the relative gene expression derived from a perfectly tetraploid population should be largely similar to the relative gene expression derived from an isogenic diploid population.

1c. Does the HRD measure contribute any significant variation?

2. The authors gave a clear rationale for classifying a tumor as being whole-genome duplicated (line 69-70), but not for the timing of WGD (line 152-154). The classification of early or late can be arbitrary. For example, a WGD event can be classified as early if it preceded the ancestor of the founding clone and late if it occurred afterwards; the timing of WGD can also be estimated based on the burden of mutations inferred to have occurred after WGD (i.e., present on one of two duplicated copies of a homologous chromosome) and those inferred to have occurred prior to WGD (present on both duplicated copies). The authors may want to clarify exactly what's their criterion for classifying WGD as early or late. Showing representative examples of early or late WGD tumors will also be helpful.

3. For the plots in Figure 4, I cannot draw any inference from the figures on the left side. For the ones on the right side, I am also unclear how the interaction is quantified, and whether the absence of interaction (e.g., in early WGD group) was due to depletion of these cell types, or suppressed interaction.

4. At the end of Introduction (line 60-64), the authors "hypothesized that WGD might promote intratumoral heterogeneity and drive unique transcriptional processes in HGSC..." I do not find any evidence of intratumoral heterogeneity in the current study. It is also likely that the establishment of a founding WGD tumor clone only progresses to tumors under an immunosuppressive environment, rather than directly causes transcriptional changes to create an immunosuppressive environment.

Given that the authors only studied late-stage cancers, it is more accurate to phrase as the transcriptional changes as "associated" with WGD, rather than driven by WGD.

Response to Reviewer Comments

Reviewer #1, expertise in ovarian cancer genomics and bioinformatics

Major points:

1. Can the authors further clarify their thoughts and reasoning for removing gene specific copy number effects from the differential gene expression analysis and contrast their findings to results obtained to a similar model not taking gene-specific copy number log ratios into account (i.e. include genes that were upregulated as a result of e.g. focal amplifications)? Various studies have now shown that copy number events are not perfectly correlated with gene expression, and gene expression driven or altered by copy number alterations might still present and reveal valuable vulnerabilities associated with WGD. Would the authors find more upregulated genes or more upregulated hallmark pathways if copy number log ratio was removed as a covariate from the applied model? And would MHC-II related genes still be significantly down-regulated? At a minimum, please justify the reasoning for removing this copy number from this analysis in more (data-driven) detail.

We thank the reviewer for this question, as it is one we had considered at length for the reasons you have described. Given that pathogenic SCNAs are likely to be gene dosage sensitive and therefore may affect the level of transcription (Fehrmann Nat Gen 2015 doi:10.1038/ng.3173; Rice Nat Comms 2017 doi.org/10.1038/ncomms14366), we felt it was more important to account for this than not to. Additionally, we constructed our model such that it was similar to the one utilized by Quinton et al (Nature 2021 doi.org/10.1038/s41586-020-03133-3), who used copy number as a covariate to identify genetic vulnerabilities in WGD tumors. We also ran our model without the inclusion of copy number log ratio, and found that the results were similar. For example, *C/ITA* was significantly downregulated in tumors with WGD across both cohorts (coefficient estimate -1.40, $p = 0.009$), as were 7 of the MHC class II genes.

In the original manuscript we had shown that expression of MHC-II genes was lower regardless of copy number status in Supplementary Fig. 1c as boxplots of gene expression. We have now also included the results of this analysis in Supplementary Table 3 and indicated this in the text.

2. Line 106-107 – Please include this data as plot/table in the manuscript, potentially as supplementary figure. (See other examples of this below. In my opinion all results/conclusions and statistics should be supported by data/figures in the manuscript).

As suggested, we have now included this data as Supplementary Fig. 1b,c. Additional information has also been added in response to comments from Reviewer 2.

3. Methylation data is also available from TCGA. Would it be possible to utilise this from the TCGA ovarian cancer cohort in addition to the methylation array data from references

6 and 19, to investigate in more detail the methylation status of the mentioned four main promoters? Is *CIITA* more frequently hypermethylated in WGD tumours in this dataset? Including this analysis might be very relevant and interesting, especially since this is further discussed as a potentially targetable vulnerability in the discussion (line 364-367)

As we did not find evidence of *CIITA* hypermethylation in the ICGC dataset we originally did not look at methylation in the TCGA data. We have now reviewed this data. The methylation arrays used in TCGA are an older methodology and only include 2 probes which relate to *CIITA*, neither of which reach the beta value cut-off for hypermethylation of 0.8. We have added this information to the manuscript.

4. Line 135 – Include figure or table to show correlation between *IRF-1* and *CIITA*. It might also be worth to briefly clarify whether *IRF1* is a positive or negative regulator of *CIITA* in the main text.

These plots have been added as Supplementary Fig.2a,b and we have added that *IRF1* is a positive regulator of *CIITA* to the text.

5. Line 190-197. The hypothesis that MHC-II expression might result in poorer PFS is not directly addressed in the presented analysis and figures. Please either re-word the hypothesis or re-do the analysis looking at MHC-II expression and PFS/OS instead. It might also be worth to perform a coxPH analysis taking multiple covariates into account, including WGD/HRD status, stage, age etc.

We have reviewed the wording and have modified it to: “We hypothesized then that reduced MHC-II expression might contribute to immune evasion in patients with tumours which have undergone early WGD, and that therefore patients with early WGD might have poorer survival outcomes.”

We have also performed a multivariate analysis using Cox proportional hazards taking into account age and stage; timing of WGD remained significant and this result has been included in the text and Supplementary Fig. 2e,f.

6. Line 206 – “data not shown”. Again, please include data and/or figures.

This data has now been included as Supplementary Tables 5 and 6.

7. Line 212 – IHC data. Please specify how this was quantified? Could IHC staining be measured using computational image analysis (such as the Halo software) to (potentially more robustly) quantify MHC-II staining/intensity? Please specify how (and from what measurements/quantifications) the p-values were derived. Please also include both of the above in the relevant Methods section, and figure legend. In addition, which figure is this

statement and the p-values (in line 215) referring to? If this is referring to Figure 2c please include the figure reference here.

Staining was scored by a trained gynaecological pathologist. The parameters scored are now included in the Methods section, which we adopted since there is no international/standardised guidelines for reporting on MHC-II expression in ovarian cancer as a diagnostic test, and the quantification method (intensity) has now been made explicit in the Results. P values were calculated on the intensity of staining from scored cores using a generalized linear mixed model to account for the fact that some cases had 2 cores and some only had 1. P values have been joined using the emmeans package for more concise reporting, and adjusted for multiple testing, and are included in full in Supplementary Fig. 3.

Figures were not originally included for line 215, as glmms are calculated separate to plotting, and we felt that simply including the none (0) vs any (1-3) staining (Fig 2c) was visually easier to appreciate, however this is now included as Supplementary Fig.3. The chi-squared test refers to Fig.2c, which has also been clarified in the text.

We have used HALO to quantify the percentage of total cells within each core that had MHC-II staining and found similar results, with fewer positive cells in early WGD samples (figure below, joint $p = 0.07$, glmm). However, as this analysis is not able to reliably distinguish which cells are expressing MHC-II (tumor vs lymphocytes vs stroma) whereas our pathologist's analysis did, we felt that reporting HALO data did not add to the manuscript.

8. Line 222-223 – mentioning 15 patient samples to be sequenced but data for only 14 samples is shown. The following sentence also only mentions 14 patients. Did one fail? If so why? Correct numbers in main text and methods section, and/or clarify if and why one sample may have failed or was excluded from the analyses.

In the original manuscript, we stated that 15 patients were selected but only 14 passed QC. To make this clearer, we have now moved the line 'For one sample, sufficient nuclei for input could not be extracted at the lysis step and was not processed further' from the Methods to the Results. The numbers are correct.

9. Line 244 and 246– Please include figure/data to support these observations?

We have now added a depiction of these results as Fig. 4a (related to proportion of cancer cells per patient expressing *CIITA*) and Fig. 4b (expression of *CIITA* in all cancer cells), and updated the text to “Comparison of the level of *CIITA* expression across all cancer cell clusters confirmed that *CIITA* was also significantly more lowly expressed in early WGD tumors (coeff. est. = 4.91 late WGD, p value <0.001, coeff. est. = 4.18 no WGD, p value <0.001, glm; Fig.4b)”. We have removed the 2nd part of the sentence (“While *CIITA* was also more lowly expressed in macrophages from cancers with early WGD, the magnitude of this was modest and may not have a biological effect”) in favour of a cleaner statement about the proportion of immune cells under the heading ‘The tumor microenvironment and its interactions with cancer cells’.

10. Line 255-257 – Please show data/figure for this in supplement.

This is now shown in Supplementary Fig. 5a. As sc/snRNAseq suffers from a high dropout rate which both limits conventional correlation analyses and visualization due to the high number of zeroes, we have now updated our manuscript with the efficient gene correlation method CS-CORE that addresses this and depicted this result in Supplementary Fig. 5a to address this and a separate comment requesting correlations with MHC-II genes.

11. Line 296-299 – NRG, Collagen and Laminin seem to be the top 3-4 hits for tumours from all three WGD categories. Is the observation described here significant? Of note also that the scales of the grey row bars (x-axis) are significantly different (>10-fold) between the three plots. Could the authors please comment on this and its meaning in the main text and/or figure legend?

Taking both Reviewer 1 and Reviewer 2’s comments into consideration, we elected to remove this paragraph, as it did not fit with the overall message and direct statistical comparison cannot be performed using CellChat, hence the observation was of limited value. The associated figure has also been removed.

12. In the discussion, the authors acknowledge limitations in current methods for assessing MHC-II specific neoantigen expression. However, it might be worth to estimate total neoantigen load (number of neoantigen candidates identified using tools such as pVacSeq) and compare the neoantigen load to MHC-II expression and WGD status in HGSC. I believe the TCGA ovarian cancer cohort should provide sufficient data for this

analysis, and the results would be very interesting in the context of the main manuscript findings and the discussion points in lines 327-335.

We applied pVacSeq to the ICGC discovery cohort and did not find a significant difference in either MHC-II or total neoantigen burden between the WGD timing groups. MHC-II neoantigens also did not correlate with individual HLA class II gene expression. We therefore did not apply this to the TCGA dataset as this was our validation cohort.

In exploring how to best address this we received feedback from multiple sources that this and similar bioinformatic tools are unreliable for MHC-II neoantigen prediction. This is in line with published data and does not appear to have been sufficiently addressed in available tools with our available data (Chen 2019, <https://doi.org/10.1038/s41587-019-0280-2>; Xie 2023 <https://doi.org/10.1038/s41392-022-01270-x>). Considering all of this, we felt that reporting a negative finding on potentially unreliable methodology was unhelpful and would neither strengthen nor weaken our findings, hence we chose instead to comment on this limitation in the Discussion as mentioned.

Minor points:

1. Line 56 “and therefore survival for almost 20 years” – could the authors reword this sentence to avoid confusion. What do the 20 years refer to, the survival or evidence?

This referred to the evidence that immune composition is prognostic (Zhang 2003 NEJM doi.org/10.1056/nejmoa020177). Given that this wording was confusing, the qualifier here has been removed, and it now reads: “New treatments have been sought for HGSC, however the response to immune checkpoint inhibitor therapy (ICI) has been poor, despite evidence that the immune milieu has an important role in ovarian cancer control and therefore survival”.

2. The use of the testing (ICGC) and validation (TCGA) datasets seem slightly inconsistent throughout the first part of the manuscript (results shown in Fig. 1 – Fig. 2a). Could the authors sign-post and justify in the text more clearly which dataset was used when and why (see some additional individual the minor comments) and potentially include results from both datasets when only one is shown in supplementary figure? More specifically, Fig 1a only shows ICGC samples (could include TCGA samples here as well); Fig 1b shows both datasets separately; Fig 1c shows both combined/integrated data; Fig 2a shows ICGC only; Fig 2b again shows both combined/integrated data). For example, could the authors justify the reasoning for integrating the two datasets in line 138-141. I am assuming that this is due to statistical power and sample numbers, but this should still be mentioned in the main text/figure legend.

As mentioned the ICGC cohort was our primary discovery dataset, and we sought to validate positive findings in the TCGA data. Since the intention of the validation cohort was to independently test the discovery results, we have where possible reported the datasets

separately, for example we left the values reported separately in Fig 1b. To address the reviewers points:

- The TCGA cohort has been added to Fig. 1a.
- Fig. 1c is data from the ActivePathways tool, which specifically integrates results from multiple independent datasets to find the statistically significant overlapping pathways. Therefore these results are never derived separately and are only reported as combined; separate analyses were not conducted with ActivePathways. We have amended the wording (lines 138-141 of the original main text) to read: “Pathway enrichment analysis was conducted using the most highly and lowly expressed genes from the DGE results for the ICGC discovery and TCGA validation cohorts using ActivePathways (35), which integrates results from multiple datasets, to identify significantly enriched Hallmark pathways.”
- Fig. 2b is indeed combined data, in order to generate sufficient power to detect a PFS/OS difference. We have updated the text to read “In order to have sufficient power to detect survival differences by WGD timing, we examined patient survival in a combined cohort (79 ICGC cases and 166 TCGA cases, plus 107 additional 107 TCGA cases for which WGS and survival information were accessible; total of 352 patients).”

We have reviewed and updated the text and figure legends to ensure that it is clear at each point which data is being referred to.

3. Line 159-173 – Include TCGA results two (either as joined or separate analysis) or alternatively justify why excluded. Compare and contrast results.

Visualizing the clustering in the ICGC discovery dataset led us to the observation that there were differences by timing, which we have subsequently assessed and confirmed with more direct methods including IHC and snRNAseq. Since the Hallmark pathways do not specifically assess the focus of our manuscript (*CIITA* and MHC-II expression), we saw limited value in adding additional analysis using this indirect method. Being conscious of a large number of analyses conducted we tried to streamline the focus of the manuscript for the reader as much as possible and prefer instead to focus on the analyses we performed to directly test this hypothesis, however we have now included a brief reference to this “This pattern of enrichment was similar in the TCGA dataset but was not statistically significant ($p = 0.14$, Chi squared test...).”, and included it as Supplementary Table 4.

4. Line 145 – very briefly describe what the Quinton et al study is if mentioning it explicitly in the main text.

This has been updated to: “Downregulated pathways were similar to those described in a pan-cancer analysis of differentially expressed pathways between cancers with and without WGD, which found Allograft rejection, Inflammatory response and Interferon gamma response to be downregulated in samples with WGD”.

5. Line 242 – “even after accounting for patient-specific differences” – please specify what these differences were and how they were accounted for.

This referred to the inclusion of patient (SampleID) as a covariate in the statistical modelling, to account for repeated measures (in this case cells) from the same patient. The wording has been simplified and updated to: “Comparison of the level of *C/ITA* expression across all cancer cell clusters confirmed that *C/ITA* was also significantly more lowly expressed in early WGD tumors (coeff. est. = 4.91 late WGD, p value <0.001, coeff. est. = 4.18 no WGD, p value <0.001, glmm; Fig. 4b). “

6. Figure 3c – lower panel. Could the y-axis be changed/transformed to sqrt of log1p scale, and all data points be shown (instead of just outliers)? Or alternatively show violin plots so the distribution of the data becomes visible.

This data is now shown in Fig. 4e, and has been replotted using re-processed snRNAseq data based on a separate request from Reviewer 2. We used the Normalize function instead of SCTransform in Seurat, which creates easier to visualize values, and it is now also shown as violin plots overlaying boxplots to more clearly show the distribution of the data. Cancer cell numbers have also been changed to proportions (but labelled with absolute total count) in the accompanying barplot so that this is also easier to visualize.

7. Figure 3d – If possible please don’t use the same colour for the overall cell count as is used for the no_WGD group since this might be confusing/misleading. Could the colour be changed to e.g. grey? Could the authors also explain the discrepancy of the number of cells shown in the top cell counts to the bottom cell counts? If there are cells that were unassigned, could this be included? Also, would it be possible to change the lower cell count (by cell type) to proportions (i.e. y-axis of 0 to 1) to make the difference in cell type abundance more clear and visible, especially since the cell count is already indicated in the top barplot?

While there are explanations for the apparent discrepancies pointed out (eg. Count differences are cancer cells at top, non-cancer cells at bottom, and are patients not clusters, which were seen in Fig. 3c and could be confused), on reflection we agree that Fig. 3d was a confusing and not especially contributory plot, since the same information is contained in other ways in the manuscript. This has therefore been removed in favor of including other, more pertinent information.

Reviewer #2, expertise in single nuclei RNA sequencing and cancer (Remarks to the Author):

Major Concerns:

In general, statistics are lacking for many of the reported findings. Importantly, for the single cell data comparisons should be made at the per-patient level and corresponding statistical test results provided for each comparison. It does not appear that any type of correction for multiple testing has been performed and considering the very large number of statistical tests performed, some method of correction is needed.

The single nuclei data statistical comparisons have been revised using either a summary per patient (eg. Proportion of cells expressing *C/ITA* per patient) with Kruskal-Wallis testing, or using mixed modelling to account for the repeated sampling of a patient within the test, as in the case of *C/ITA* expression.

Regarding multiple testing correction, this is not required where mixed modelling has been used as this is encompassed internally. The exceptions were the DGE analysis on bulk RNAseq data where individual per-gene models were conducted rather than the dataset as a whole (multiple testing correction had already been performed on this data and described in the methods - original manuscript line 469), and in the IHC intensity comparisons which were conducted for location individually – this has been updated.

We have also carefully reviewed the manuscript to assess for any additional areas where multiple testing correction was required and have now included this, for example where per gene Kruskal-Wallis tests are used.

There is also no description in the methods showing clearly for each type of test what the significance threshold was.

We have specified the significance threshold, using the conventional threshold of $p \leq 0.5$, in the Methods section under 'Statistics'.

Cancer cells should be reclustered and analyzed separately for Figure 3.

This has been done, please see Fig. 3c,d. This data was reprocessed using the Normalize function in Seurat instead of SCTransform (since cells had already been annotated as cancer cells, which we feel to be the major strength of SCTransform). This yielded a data structure which is easier for visualization and statistics, and this was therefore also used for downstream analyses, for example comparison of *C/ITA* expression.

Minor Concerns:

The introduction could include more discussion on what is known about MHC II expression by cancer cells and the consequences for immune recognition and response to checkpoint blockade.

Additional discussion has been added to the introduction: "While immune escape related to reduced neoantigen presentation has been largely described with regard to MHC-I in HGSC and

other cancer types (17-19), there is increasing suggestion that MHC-II expression also plays a prominent role (20). Canonical antigen presenting cells, including macrophages, dendritic cells and B cells, as well as cancer cells can express MHC-II and present neoantigens (20, 21). The presence of MHC-II on cancer cells can even predict response to immune checkpoint inhibitors (ICI).”

Page 2, line 44 – it’s unclear just what is meant here. If the use of “continues to be acquired” refers to an increase over disease progression as a result of comparing primary tumors with end stage, then those analyses should be done, and the results presented. Otherwise this sentence should be modified or removed.

We have modified the wording here to be clearer: ‘WGD has been observed in 50-60% of primary ovarian cancers, yet we found a higher proportion in our study of end-stage homologous recombination (HR) deficient HGSC (79.6% of tumors), suggesting that WGD continues to be acquired after diagnosis’.

Page 2 lines 80-81. What is the significance threshold used to determine the number of DEGs? This is a very large number of DEGs for an experiment of this type. A full list of DEGs needs to be provided.

The significance threshold to identify DEGs was $p < 0.05$. A list of the 689 differentially expressed genes that were statistically significant across both the discovery and validation cohorts at any magnitude of change was already provided as Supplementary Table 2, and in response to Reviewer 1 we have also added the 550 DEGs that were identified in the analysis without copy number log ratio as a covariate as Supplementary Table 3.

We had endeavored to be precise in the breakdown of numbers reported, but acknowledge that it is perhaps confusing and unnecessary to report on the number of genes which were statistically significant at any magnitude from the discovery cohort (this is the large 7,534 which was questioned). We have therefore reworded our text to only detail those in the discovery cohort which were statistically significant and met our magnitude threshold: “Of the 16,375 genes input into the model, 82 were significantly upregulated and 593 significantly downregulated using a coefficient estimate threshold of ± 1.5 and a p value < 0.05 .”

We can increase the number of genes in this Supplementary Table to include those which were significant in one cohort only, however many have such a small magnitude of difference, despite being statistically significant we felt they were unlikely to be biologically meaningful.

Page 3, lines 82-86. A presentation of the results of the purity analysis should be provided as tables and figures prior to the DEG analysis so that the potential of the covariates can be assessed by the reader.

This was previously stated in the Methods but has now been moved to the main text (“median tumor purity estimated by FACETS was 68.7% in tumors without WGD and 62% in tumors with WGD”) and more detail has been included in Supplementary Table 1, preceding the DGE analysis as suggested.

Page 4, lines 100-112. Please split the cohort into 4 groups: WGD+ with loss of Chr 6, WGD- with no loss of chr 6, WGD- with loss of Chr 6, WGD- with no loss of Chr 6, and then show expression of these genes in each of these 4 groups. Visualizing these values would be very helpful.

Patients with WGD do not have loss of the loci in 6p which contain these genes. We have confirmed this manually and it was already visualized in Supplementary Fig.1a. In response to this comment and a comment from Reviewer 1 we have added the plot of coefficient estimates from the DGE for the segments containing MHC class II genes (Supplementary Fig. 1c), as well as a comparison of segment means in MHC II genes versus other genes in the same region, showing that there is no difference (Supplementary Fig. 1b), in support of our assertion that this is not a copy number driven effect. Lastly, unadjusted gene expression values for genes of interest are already shown in Supplementary Fig. 2d (Supplementary Fig. 1c in original manuscript).

Page 4 lines 116-117. 166 HGSC tumors is a small subset of TCGA – how were these tumors selected in?

These were the subset of primary solid tumor samples accessible using TCGABiolinks for which all of the required information could be accessed: transcriptomics data and copy number data, with matched tumor purity from Aran et al. and copy number signature abundance data available from Steele et al.

Page 5, line 139. Please show the full results of pathway analysis in ICGC and TCGA separately, and then when combined.

We have included the full results of the integrated pathway analysis by ActivePathways in the manuscript. ActivePathways takes as its input a list of genes (it would be the user’s choice whether these are up or downregulated, or otherwise important). Its strength is its integrative capability to take multiple datasets (ie. in our case a list of DGEs from the discovery and validation cohorts) and internally generate one final result of statistically significant pathways across both cohorts using a ranked hypergeometric test. We have reviewed the language in our manuscript to ensure that this process is clear, and it now reads “Pathway enrichment analysis was conducted using the most highly and lowly expressed genes from the DGE results for the ICGC discovery and TCGA validation cohorts using ActivePathways (35), which integrates results from multiple datasets, to identify significantly enriched Hallmark pathways”.

Please show the comparison of the Dewhurst and previously published PCAWG WGD estimation so that the reader can assess if the two methods in fact had good concordance by indicating samples profiled with both methods.

This was included in our original submission as Supplementary Fig. 1b. In the updated manuscript it is Supplementary Fig. 2c.

Page 5, lines 165-173: Recent single cell data have shown the molecular subtypes previously reported (and this citation is not the originating paper for the molecular subtypes anyway), are largely driven by sample composition, and are reflective of stroma content in the tumor, which is where immune cells reside in HGSC.

As succinctly summarized by Chen and colleagues in their development of a consensus subtype classifier (2018 Clin Cancer Res; [10.1158/1078-0432.CCR-18-0784](https://doi.org/10.1158/1078-0432.CCR-18-0784)) “initial large-scale efforts to classify HGSOC of the ovary did not reveal any reproducible subtypes. Tothill and colleagues reported four distinct HGSOC subtypes”, and the Tothill findings were published in 2008 prior to similar findings in the TCGA study in 2011. We have referenced both the Tothill and Chen papers, however we are open to including other references.

It was recognized early on in both the Tothill and TCGA papers that fibroblasts (C1/mes) and lymphocytes (C2/imm) were major features of two of the subtypes but as the Reviewer points out this has become clearer in single cell analyses. That the transcriptional subtypes determined in bulk RNAseq data reflect microenvironment contribution doesn't alter the association of C5 with WGD noted here.

Many studies that have detected TILs in the epithelial compartment of the tumour (eg Zhang 2003 NEJM 10.1056/NEJMoa020177; OTTA Consortium 2017 JAMA Oncology 10.1001/jamaoncol.2017.3290; Garsed 2018 Clin Cancer Res 10.1158/1078-0432.CCR-17-1621). We have added a clarifier - “epithelial and stromal immune cell infiltration “ – to ensure this is clear.

Figure 2: Please also provide a heatmap of gene expression data showing the top 1000 variable genes across the samples.

We have produced this heatmap of 1000 genes with the highest median absolute deviation across all samples for the ICGC data, shown below. We have not included this in the manuscript figures as it does not show any additional findings or add any value to the current manuscript with respect to our findings regarding WGD or MHC-II gene expression.

Page 7 lines 189-192: This analysis should be restricted to the 116 TCGA tumors in analysis previously in this report.

These lines appear to refer to a section of text that is slightly between analyses, however we have taken this to refer to the survival analysis just below. While we agree that using the exact same cases for all analyses would have been preferable, we conducted a sample size power analysis prior to performing the survival analysis, and found that for a medium effect size a minimum of 53 individuals in each group was required. Therefore, neither the ICGC or TCGA cohorts individually or combined were large enough. As the survival analysis did not require matched RNAseq or tumor purity data, we were able to include more TCGA cases and improve our power to detect survival differences.

We have modified the wording in the text to be clear on why we took this approach: “In order to have sufficient power to detect survival differences by WGD timing, we examined patient survival in a combined cohort (79 ICGC cases and 166 TCGA cases, plus 107 additional TCGA cases for which WGS and survival information were accessible; total of 352 patients)”.

Page 7, lines 210-217: Please include the complete set of CIBERSORT annotations. A heatmap with corresponding table showing cell types and quantifications across the samples should be included.

We have now included this data as Supplementary Table 6. We have also included the CIBERSORT statistical analysis summary in Supplementary Table 5. As a large number of figures have been added to address the reviewer's comments as a whole, we have prioritised including figures directly related to assessing our hypothesis and hope that including this data in this format is acceptable.

Show dot plot of CIITA, MHC expression by cluster in 3C

We were not entirely certain what plot is being requested, as 'MHC expression' refers to multiple genes. The reclustered cancer cell plot of *CIITA* expression is now Fig. 4b, and we have shown expression of individual MHC-II genes which were statistically significant in the bulk RNAseq model as Supplementary Figure 4. We hope that we have interpreted this correctly and that this addresses the Reviewer's comment.

Boxplots showing proportion of total cells for each cell type should replace 3d. Statistical analysis should be applied to show proportions are different.

In response to this and comments from other reviewers, we felt this figure was confusing and it has now been removed. However, this point is also addressed by Fig. 4a and Supplementary Fig. 5b, showing aggregate proportion of cancer cells and immune cells respectively, expressing *CIITA* per patient and using a Kruskal-Wallis test to test per patient across WGD timing category. Overall proportions of cell subsets are also provided in Supplementary Table 8.

Please include why 15 samples were used for scRNA-Seq but data is only shown for 14 samples.

In the original manuscript, we stated that 15 patients were selected but only 14 passed QC in the Results. To make this clearer, we have now moved the line 'For one sample, sufficient nuclei for input could not be extracted at the lysis step and was not processed further' from the original manuscript's Methods to the Results.

What is the cell level correlation of CIITA and MHC? Cluster comparisons are made but what about on the cell level. Does low CIITA correlate with low MHC II across tumor cells?

We have added this information to the manuscript as a Supplementary Fig. 5a, since there are multiple MHC-II genes, in the interest of encompassing these succinctly. We have added the text: “All MHC-II genes which were statistically significant in the bulk RNAseq DGE had a statistically significant correlation with *CIITA* expression within cancer cells (Supplementary Fig. 5a)”.

Figure 3: Given previous publications showed that tumor cell clusters in HGSC are typically from a single patient please provide Figure 3a with cells labelled with Patient ID as a main figure and not supplemental. A discussion of why this may be different in your report should be included in this manuscript.

As requested we have reclustered cancer cell clusters and did so using the Normalize function in Seurat (instead of SCTransform which we feel performs best to classify cells but is more problematic when comparing gene expression due to its continuous but binned structure). Using the reclustered data, substantially less overlap can be seen. This is seen in Fig. 3c.

Aside from this, there are several reasons why our data might be more likely to overlap. First, while other cohorts include samples which are likely to have more mixed cell types (eg. Ascites, solid tumour from ovary, bowel, etc; Vazquez-Garcia 2022 doi.org/10.1038/s41586-022-05496-1) or from different timepoints (Nath 2021 doi.org/10.1038/s41467-021-23171-3), ours were all primary solid tumour samples, and 10 of these were ovary. We have included this information as a Supplementary Table 6.

Lastly, we note that there are multiple examples of other studies where tumour cells have some overlap (eg. Vazquez-Garcia 2022; Izar 2020, doi.org/10.1038/s41591-020-0926-0; Nath 2021), hence we feel these are insubstantial differences of our data, and have provided additional information for the interested reader.

It appears as though there are significant numbers of non-tumor cells with WGD. Is this an artifact of 2D visualization? Can a third dimension be added to these plots to help clarify? Particularly for cells like CD8 T cells, endothelial cells and fibroblasts.

We agree that the figure and text were unclear - the bottom UMAP in the original manuscript Fig. 3a shows all cells from each patient, including non-cancer cells, colored by the WGD timing category derived from bulk WGS. This does not imply that non-cancer cells have WGD, just that the tumour as a whole is classified this way. Combining this comment with the earlier comment about Figure 3, this has been addressed by reclustering cells and showing the 3rd UMAP clearly annotated as reclustered cancer cells only in Fig. 3d.

Page 10, lines 248-251. Please provide a UMAP showing the location of cancer cells expressing *CIITA*.

This was previously already provided in Fig. 3b. This is now Fig. 4b in the revised manuscript.

There are no statistics for Figure 4. If the authors are claiming cell chat predicted MHC signaling is greater in the no or late WGD cells then they need to show statistics comparing per patient levels of this signal.

CellChat only visualizes statistically significant interactions, which was not specifically stated in the original manuscript as an error of omission. However, CellChat does not have the capability to perform a statistical comparison between 2 pathways, in this case MHC-II signaling. We considered whether it would be truthful/useful to apply statistics manually to the CellChat output, but concluded that this would be arbitrary and not a meaningful statistical analysis. Reviewing others' use of CellChat, including in manuscripts published in Nature Communications, others have also taken a similar approach and not applied post-hoc statistical analyses.

We feel that this is a useful descriptive analysis as an adjunct to the analyses where we have applied rigorous statistics, however agree that it is important to make it clear that this is not a statistical comparison. We have therefore changed the wording to: "In a descriptive analysis of interactions between cell subsets in the different WGD categories, tumors with early WGD had markedly low MHC-II signaling (Fig. 5). Only macrophages displayed any MHC-II signaling in tumors with early WGD; in contrast, extensive intercellular MHC-II signaling between macrophages and both cancer and non-malignant cells was observed in tumors with late or no WGD."

Statistics need to be provided for both figure panels in 5 to show on a patient level the reported differences are significant.

This analysis and figure have been modified to include statistical tests with p values (now Fig. 4c,d).

Were the samples in this cohort of the same stage and grade?

All samples for snRNAseq had been diagnosed with HGSOC (grade 2 or 3) of advanced stage (II/IV). This information has now been included as a Supplementary Table 7.

There is no indication which of these samples are HRD vs HRP, and this is an important factor in the TME of HGSC. Please provide the HR status of each tumor.

Information on *BRCA1/2* status and HRDsum scores have been included in Supplementary Table 7.

There is no data availability statement.

The data availability statement was previously on line 552 in the original manuscript, with the Methods. This has now been moved to its own heading.

Reviewer #4, expertise in whole genome duplication, cancer genomics and chromosomal rearrangements (Remarks to the Author):

1. The authors discussed the caveats of performing differential gene expression analysis for tumors with large segmental copy number changes.

1a. In the general linear model shown in Fig. 1a and also in method, is the "Norm. HTseq value" on the left hand side the log-transformed read count? It does not make sense to me if this is the normalized count (without log transformation).

We agree that this is confusing and thank the Reviewer for pointing this out. The normalized HTseq values were the raw input before transformation, but the values used for input into linear modelling are the log-transformed counts from `limma::voom`. We have amended this in the figure. This was stated in the methods "Limma::voom (RRID:SCR_010943) v3.50.3 (57) was used to generate normalised and log-transformed values for input into differential gene expression", but was contradicted by the figure.

1b. What was being used in the copy number logR? I would take this to be the normalized DNA copy number, i.e., segmental copy number divided by ploidy. From line 83-86, the authors mentioned that "by definition tumors with WGD will have higher total copy number per gene." I do not think this matters as both gene expression (l.h.s. of the linear model) and DNA copy number (r.h.s. of the linear model) are relative measurements. As a simple example, the relative gene expression derived from a perfectly tetraploid population should be largely similar to the relative gene expression derived from an isogenic diploid population.

The Reviewer is correct in their assumption of definition of copy number logR (CNLR). The answer to this overlaps heavily with the first question from Reviewer 1 and we paraphrase this response here:

Whether or not to include some kind of covariate to account for copy number is one we considered at length. Given that pathogenic SCNAs are likely to be gene dosage sensitive and therefore may affect the level of transcription (Fehrmann Nat Gen 2015 doi: 10.1038/ng.3173; Rice Nat Comms 2017 doi.org/10.1038/ncomms14366), we felt it was more important to account for this than not to. Additionally, we constructed our model such that it was similar to the one utilized by Quinton et al (Nature 2021 doi.org/10.1038/s41586-020-03133-3), who used copy number as a covariate to identify genetic vulnerabilities in WGD tumors. We also ran our model without copy number log ratio, and found that the results were similar. For example, *C/ITA* was significantly downregulated in tumors with WGD across both cohorts (coefficient estimate -1.40, $p = 0.009$), as were 7 of the MHC class II genes.

1c. Does the HRD measure contribute any significant variation?

We tested several different HRD covariates and their effect on the model. Model quality assessed by Akaike information criteria using no HRD covariate, *BRCA1/2* mutation status, CHORD, SBS3, HRDsum and copy number signature 17 (CN17) demonstrated that all models had very similar AIC values, indicating that there was minor differences only. Only *BRCA1/2* mutation status and CHORD performed significantly better using ANOVA comparison, but the benefit was modest. As both of these require controlled data from TCGA we elected to use CN17 as our covariate.

2. The authors gave a clear rationale for classifying a tumor as being whole-genome duplicated (line 69-70), but not for the timing of WGD (line 152-154). The classification of early or late can be arbitrary. For example, a WGD event can be classified as early if it preceded the ancestor of the founding clone and late if it occurred afterwards; the timing of WGD can also be estimated based on the burden of mutations inferred to have occurred after WGD (i.e., present on one of two duplicated copies of a homologous chromosome) and those inferred to have occurred prior to WGD (present on both duplicated copies). The authors may want to clarify exactly what's their criterion for classifying WGD as early or late. Showing representative examples of early or late WGD tumors will also be helpful.

We described how timing of WGD was deduced in the original manuscript lines 151-154. We also included a figure comparing this heuristic method to the consensus method used by Gerstung et al (Supplementary Fig. 1b in original manuscript, and Supplementary Fig. 2c in revised manuscript), showing good concordance. For additional clarity, we have now added the following wording: "Their method uses genomic regions with a total copy number of 2 to categorise tumors with more heterozygous regions as having undergone genome duplication before the majority of losses (early), and tumors with more homozygous regions classified as having undergone WGD after the majority of losses (late). While this is acknowledged to be heuristic, our findings were concordant with the previous analysis by Gerstung et al. (2) for the 37 ICGC samples with WGD analysed by both methods (Supplementary Fig. 2c)."

3. For the plots in Figure 4, I cannot draw any inference from the figures on the left side. For the ones on the right side, I am also unclear how the interaction is quantified, and whether the absence of interaction (e.g., in early WGD group) was due to depletion of these cell types, or suppressed interaction.

The plots on the left of the figure visualizes statistically significant interactions, and has been included primarily as a demonstration of the breadth of all interactions between cell clusters; as this is more methodological in nature we have moved this to Supplementary Fig. 5c. The MHC-II signaling plot (now Fig. 5) visualizes the absence or presence of statistically significant interactions between cell clusters. It does not give information on the strength of the interaction. Additional contextual information has been added to the figure legend.

While this analysis is descriptive only (not directly comparative), we feel it has value and has been clearly marked in the text as a descriptive analysis: “In a descriptive analysis of interactions between cell subsets in the different WGD categories, tumors with early WGD had markedly low MHC-II signalling (Fig. 5). Only macrophages displayed any MHC-II signalling in tumors with early WGD; in contrast, extensive intercellular MHC-II signalling between macrophages and both cancer and non-malignant cells was observed in tumors with late or no WGD.”.

Regarding whether any differences are due to suppressed interaction or depletion, this cannot be inferred from this data, and would be the scope of further experiments.

4. At the end of Introduction (line 60-64), the authors "hypothesized that WGD might promote intratumoral heterogeneity and drive unique transcriptional processes in HGSC..." I do not find any evidence of intratumoral heterogeneity in the current study.

This was a comment introducing the broad hypothesized consequences of WGD. As we have not specifically assessed this, we have removed ‘might promote intratumoral heterogeneity’, so that it now reads: “We hypothesized that WGD might drive unique transcriptional processes in HGSC that may contribute to disease recurrence and treatment resistance.”

It is also likely that the establishment of a founding WGD tumor clone only progresses to tumors under an immunosuppressive environment, rather than directly causes transcriptional changes to create an immunosuppressive environment. Given that the authors only studied late-stage cancers, it is more accurate to phrase as the transcriptional changes as "associated" with WGD, rather than driven by WGD.

Our study exclusively studies samples taken at primary diagnosis, as stated in the first sentence of the Results, but we agree that care should be taken not to ascribe causation where only an association has been demonstrated. We have updated this wording to: “We discovered that MHC-II expression is lowest in tumors which have acquired WGD early in tumor evolution, and further demonstrated reduced MHC-II expression in subsets of tumor cells rather than in canonical antigen-presenting cells”.

Reviewers' Comments:

Reviewer #1:

Remarks to the Author:

I thank the authors for their careful and thoughtful responses to my comments. The authors have now addressed all of my questions satisfactorily, have clarified the text, and added further information, analyses and figures to support the conclusions of their study.

Reviewer #2:

Remarks to the Author:

The authors have addressed all concerns.

Reviewer #4:

Remarks to the Author:

No more comments as the authors have adequately addressed my comments.